# PAPM: A Physics-aware Proxy Model for Process Systems

## Abstract

Process systems, which play a fundamental role in various scientific and engineering fields, often rely on computational models to capture their complex temporal-spatial dynamics. However, due to limited insights into the intricate physical principles, these models can be imprecise or inapplicable, coupled with a significant computational demand exacerbating inefficiencies. To address these challenges, we propose a **p**hysics-**a**ware **p**roxy **m**odel (**PAPM**) to explicitly incorporate partial prior mechanistic knowledge, including conservation and constitutive relations. Additionally, to enhance the inductive biases about strict physical laws and broaden the applicability scope, we introduce a holistic temporal and spatial stepping method (TSSM) aligned with the distinct equation characteristics of different process systems, resulting in better out-of-sample generalization. We systematically compare state-of-the-art pure data-driven models and physics-aware models, spanning five two-dimensional non-trivial benchmarks in nine generalization tasks. Notably, PAPM achieves an average absolute performance improvement of 6.4%, while requiring fewer FLOPs, and only 1% of the parameters compared to the prior leading method, PPNN.

## 1 Introduction

Process systems, spanning molecular dynamics to turbulent flows, are foundational across diverse scientific and engineering domains (Cameron & Hangos, 2001). Computational modeling and simulation are pivotal for grasping the intricate temporal-spatial dynamics of these systems. Being central to this progress, process models encapsulate vital conservation and constitutive relations, while analytical solutions are often unattainable due to these models' inherent complex non-linear characteristics. Consequently, these models are converted into numerical solutions through spatial and temporal discretization, employing traditional solvers such as finite difference, finite volume, finite element, and spectral methods (Zachmanoglou & Thoe, 1986).

However, two main challenges complicate their applications. **First**, due to incomplete insights into associated physics, the governing constitutive relations for certain process systems remain elusive, rendering pure first-principled mechanistic models either inapplicable or imprecise. Proxy models, bridging partially known physics and observational data, have emerged as an approach to capture these dynamics (Nguyen et al., 2023). **Second**, while traditional numerical simulations offer valuable insights, they are computationally intensive, particularly in scenarios necessitating frequent model queries like reverse engineering forward simulation (Dijkstra & Luijten, 2021), optimization design (Gramacy, 2020), and uncertainty quantification (Zhu et al., 2019). These underscore the pressing need for models balancing computational efficiency with accuracy, as highlighted in recent scientific and engineering advancements (Karniadakis et al., 2021; Wang et al., 2023).

Recent advancements in scientific machine learning (SciML) have paved the way to tackle computational challenges more effectively. CNNs (Bhatnagar et al., 2019; Stachenfeld et al., 2021) and GNNs (Sanchez-Gonzalez et al., 2020; Li & Farimani, 2022) target spatial dynamics within mesh grids, while RNN (Kochkov et al., 2021) and LSTM (Zhang et al., 2020) focus on temporal progression. Neural operators (Lu et al., 2019; Li et al., 2020; Hao et al., 2023b; Gupta & Brandstetter, 2022; Raonić et al., 2023), excelling in mapping between temporal-spatial functional spaces, have demonstrated success across various types of PDEs. Notably, these methods adopt a supervised learning-from-data paradigm, emphasizing inductive biases only about network architecture over strict physical laws. However, their reliance on extensive datasets and presumed train-test unifor-

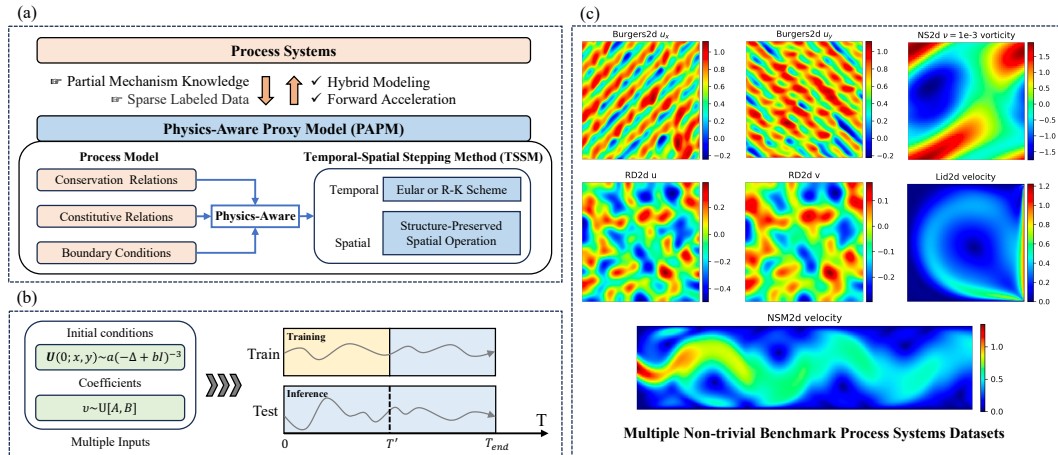

Figure 1: (a) Relationship structure diagram between process systems and PAPM. (b) Our task for time extrapolation across multiple inputs. (c) Multiple non-trivial benchmarks from sciences are provided.

mity can result in inaccuracies, especially during prolonged evolutions in out-of-sample scenarios, *e.g.*, unseen initial conditions and coefficients, such as Reynolds number, and extrapolating in time.

As a more promising strategy, physics-informed machine learning (PIML) integrates physics-based prior knowledge, such as strict physical laws, into neural networks (NNs) (Li et al., 2021; Hao et al., 2022; Meng et al., 2022). This enhances the sample efficiency and generalizability of NNs, particularly vital in scenarios with limited labeled data (Cuomo et al., 2022). The incorporation of this knowledge diverges into two main methods: **learning biases** and **inductive biases** (Karniadakis et al., 2021).

**Learning Biases.** Physics-informed neural networks (PINNs) (Raissi et al., 2019; Cai et al., 2021; Lu et al., 2021b) embed a multi-layer perceptron tailored to specific PDEs, utilizing automatic differentiation for initial and boundary conditions. However, a system with new parameters requires re-training. Additionally, its demand for comprehensive mechanism information, coupled with optimization challenges and instability under complex equations (Wang et al., 2022), limits its real-world applicability. This limitation continues with newer models like PI-DeepONet (Wang et al., 2021) and PINO (Li et al., 2021). Despite their hybrid nature, both these models require a detailed understanding of system mechanics, posing challenges in practical settings.

**Inductive Biases.** Contrary to the learning biases that integrate complete physics knowledge into its loss function, this method only leverages the data loss while explicitly incorporating either entire or partial mechanistic knowledge into the network architecture, aptly termed as "**physics-aware**" models. The purpose of such reliance is to reinforce the inductive biases concerning strict physical laws. Inspired by traditional numerical methods, recent investigations have identified links between neural network designs and equations (Long et al., 2018; Seo et al., 2020; Liu et al., 2022; Huang et al., 2023b; Akhare et al., 2023; Rao et al., 2023; Huang et al., 2023a). However, these physics-aware models are mainly spatial-derivative-focused, often overlooking integral components like conservation or constitutive relations, which leads to unreliable solutions.

Recognizing that real-world system modeling often necessitates conservation relations rooted in diffusion and convection flows, this work aims to delve deeper. Notably, different process models correspond to specific conservation or constitutive equations based on inherent system characteristics (Cameron & Hangos, 2001; Takamoto et al., 2022; Hao et al., 2023a). While all of these methods overlook this attribute, it's beneficial to identify them and select the appropriate temporal-spatial stepping modeling method for approximating unknown dynamics. Doing so can embed strict physical laws as inductive biases into the network architecture, resulting in better out-of-sample generalization.

As illustrated in Fig. 1 (a), we propose a **p**hysics-**a**ware **p**roxy **m**odel (**PAPM**), focused on out-of-sample scenarios, as shown in Fig. 1 (b). PAPM is a composite of several modules by combining partial prior knowledge and NNs, which is tailored to specific conservation and constitutive equations, encompassing a vast array of PDEs and algebraic equations. The core contributions of this work are:

- The proposal of PAPM, a novel physics-aware architecture design that explicitly incorporates partial prior mechanistic knowledge such as boundary conditions, conservation, and

constitutive relations. This design proves to be superior in terms of both training efficiency and out-of-sample generalizability.

- The introduction of TSSM, a holistic spatio-temporal stepping modeling method. It aligns with the distinct equation characteristics of different process systems by employing stepping schemes via temporal and spatial operations, whether in physical or spectral space.

- A systematic evaluation of state-of-the-art pure data-driven models alongside physics-aware models, spanning five two-dimensional non-trivial benchmarks in nine generalization tasks, as depicted in Fig. 1 (c). Notably, PAPM achieved an average absolute performance boost of 6.4%, requiring fewer FLOPs, and utilizing only 1%-10% of the parameters compared to alternative methods.

## 2 RELATED WORK

**Proxy Models.** Proxy models serve as streamlined versions of complex models, reducing computational expenses while retaining essential features of the original systems, often referred to as surrogate models (Alizadeh et al., 2020). These models are crucial in applications like digital twins (Chakraborty et al., 2021), where they compensate for the limitations of traditional models by incorporating observational data. This enhances the representation of real system dynamics and proves effective in various uses such as system analysis and complex tasks like optimization and uncertainty assessment (Zhang et al., 2021; Bahrami et al., 2022; Zhu et al., 2019).

**Pure Data-driven Method.** ConvLSTM (Shi et al., 2015) captures spatial dependencies via convolution operations while managing temporal dynamics using recurrent units. Dil-ResNet (Stachenfeld et al., 2021) is devised to forecast the difference between consecutive states, incorporating the encoder-process-decoder pattern (Sanchez-Gonzalez et al., 2020) via dilated convolutional networks (Yu et al., 2017). Another line is the neural operator. Fourier neural operator (FNO) (Li et al., 2020) learns the operator by harnessing the spectral domain alongside the Fast Fourier Transform. DeepONet (Lu et al., 2019) approximates various nonlinear operators by leveraging branch and trunk networks for input functions and query points. Building upon this, MIONet (Jin et al., 2022) addresses the challenges of multiple input functions within the DeepONet framework. Moreover, U-FNets (Gupta & Brandstetter, 2022) and convolutional neural operators (CNO) (Raonić et al., 2023) are modified U-Net (Ronneberger et al., 2015) variants, where the former replace U-Net's layers by FNO's Fourier blocks, and the latter replace them by predefined convolutional block.

**Physics-aware Method.** Inspired by finite volume, FINN (Karlbauer et al., 2022) innovatively employs flux and state kernels for modeling components of advection-diffusion equations. PPNN (Liu et al., 2022) bakes prior-knowledge terms from low-resolution data, estimates unknown parts with the trainable network, and uses the Euler time-stepping difference scheme to form a regression model for updating states. PiNDiff (Akhare et al., 2023) integrates partial physics knowledge into the NN block for a specific composite manufacturing process, ensuring mathematical integrity via differentiable programming. PeRCNN (Rao et al., 2023) employs convolutional operations to approximate unknown nonlinear terms in PDEs while incorporating known terms through difference schemes.

**Spatial and temporal decomposition.** PiMetaL (Seo et al., 2020) decomposed modeling into spatial and temporal parts, where the former shares the spatial derivatives in the global task, and the latter learns specific adaptively for individuals. NeuralStagger (Huang et al., 2023b) accelerated the solution of PDEs by spatially and temporally decomposing the original learning tasks into several coarser-resolution subtasks. Moreover, these physics-aware methods just discussed (Karlbauer et al., 2022; Liu et al., 2022; Akhare et al., 2023; Rao et al., 2023) all decomposed the spatial and temporal part, where the spatial part updates the state in each time step, and temporal relations are modeled via Neural ODE (Chen et al., 2018) or Eular schemes. These decomposition ways significantly reduce data requirements and are more conveniently integrated between prior physics with data-driven models for modeling dynamic systems.

## 3 PRELIMINARIES

This section presents the foundational description of process systems, known as the process model. Additionally, further clarification is provided on the specific problem in this work.

**Process Model.** Pivotal in engineering disciplines, process models serve to represent and predict the dynamics of diverse process systems. This model's mathematical foundation relies on two essential sets of equations: the differential conservation equations, governing the dynamic behavior of fundamental quantities, and the algebraic constitutive equations, which describe the interactions among different variables. Further details are provided in Appendix A.1.

Formally, Eq. 1 and Eq. 2 represent the conservation and constitutive equations, respectively. The designed physics-aware proxy model (**PAPM**), depicted in Fig. 2, integrates these principles by embedding various types of prior knowledge.

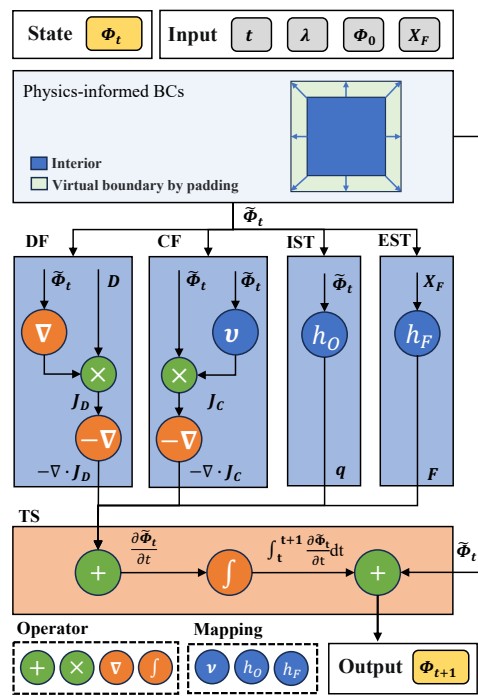

$$\begin{cases} \dfrac{\partial \Phi}{\partial t} = -\nabla \cdot (\boldsymbol{J}_C + \boldsymbol{J}_D) + \boldsymbol{q} + \boldsymbol{F} \\ \boldsymbol{J}_C = \Phi(\boldsymbol{x}, \boldsymbol{t}) \cdot \boldsymbol{v}, \quad \boldsymbol{J}_D = -\boldsymbol{D} \cdot \nabla \Phi \end{cases} \quad (1)$$

$$\begin{cases} \boldsymbol{v} = \boldsymbol{v}(\Phi), \quad \boldsymbol{D} = \lambda \\ \boldsymbol{q} = h_O(\Phi), \quad \boldsymbol{F} = h_F(\boldsymbol{X}_F) \end{cases} \quad (2)$$

where Eq. 1 comprises four essential elements: the diffusion flows $\boldsymbol{J}_D$, convection flows $\boldsymbol{J}_C$, the internal source $\boldsymbol{q}$, and the external source $\boldsymbol{F}$, with $\Phi$ denoting the physical quantity. These four elements correspond to Diffusive Flows (DF), Convective Flows (CF), Internal Source Term (IST), and External Source Term (EST) in PAPM's structure diagram, as depicted in Fig. 2, respectively.

Figure 2: A detailed structure of PAPM. The mapping can be expressed by specific constitutive equations or by NNs.

In Eq. 2, $\boldsymbol{v}$ denotes the velocity of the physical quantity being transmitted, $D$ is the diffusion coefficient. Here, $v$, $h_O$, and $h_F$ are the corresponding algebraic mapping, and this part determines whether NN is needed for learning mapping according to the specific problem. See Section 4.1 for more discussion.

**Problem Formulation.** Focus on out-of-sample scenarios containing unseen initial conditions and coefficients, and extrapolating in time. Under different ICs and coefficients, given initial step size $t_0$, which contains a short temporal trajectory $\boldsymbol{S}^0 = (\boldsymbol{s}^0, \cdots, \boldsymbol{s}^{t_0-1})$, the following trajectory $(\boldsymbol{s}^{t_0}, \cdots, \boldsymbol{s}^{T_{end}})$ should be predicted, and each $\boldsymbol{s}^t \in \mathcal{R}^d$ is a vector, which consists of dense spatial fields, like velocity, vorticity, pressure and temperature. Moreover, due to the high cost of generating labeled data, the training dataset $\mathcal{D}$ only contains the initial $T'$-step trajectory, $t_0 \leq T' \ll T_{end}$. Formally, the dataset $\mathcal{D} = \{(\boldsymbol{a}_k, \boldsymbol{S}_k)\}_{1 \leq k \leq D}$, where $\boldsymbol{S}_k = \mathcal{G}(\boldsymbol{a}_k)$, $\boldsymbol{a}_k$ contains a set of inputs, that is a set of initial trajectory $\boldsymbol{S}_k^0$ and coefficient $P_k$, while $\boldsymbol{S}_k$ contains the following trajectory $\boldsymbol{S}_k^T = (\boldsymbol{s}_k^{t_0}, \cdots, \boldsymbol{s}_k^T)$, and the mapping $\mathcal{G}(\cdot)$ is our goal to learn. For each $\boldsymbol{s}_k^t$, we discretize it on the grid $\{x_i \in \Omega\}_{1 \leq i \leq N'}$, and $s_{k,i}^t = \boldsymbol{s}_k^t(x_i)$. In a nutshell, for modeling this operator $\mathcal{G}(\cdot)$, we use a parameterized neural network $\tilde{\mathcal{G}}_\theta$, inputs $\boldsymbol{a}_k$, and outputs $\tilde{\mathcal{G}}_\theta(\boldsymbol{a}_k) = \tilde{\boldsymbol{S}}_k$, where $1 \leq k \leq D$.

Here, the training dataset is sparse, where $T = T'$, $t_0 \leq T' \ll T_{end}$, the size $D = D_0$, and $D_0$ is tiny, while the testing dataset $T = T_{end}$. PAPM $\mathcal{G}_\theta$ with parameters $\theta$ autoregressively predict $\boldsymbol{s}^t = \mathcal{G}_\theta(\boldsymbol{S}_t)$, where $\boldsymbol{S}_t$ is a trajectory with length $t_0$ form $\boldsymbol{s}^{t-t_0}$ to $\boldsymbol{s}^{t-1}$, $t_0 \leq t \leq T_{end}$. Our goal is to minimize the $L_2$ relative error loss between the prediction $\tilde{\boldsymbol{S}}_k$ and real data $\boldsymbol{S}_k$ as,

$$\min_{\theta \in \Theta} \frac{1}{D_0} \sum_{k=1}^{D_0} \frac{\| \boldsymbol{S}_k - \tilde{\boldsymbol{S}}_k \|_2}{\| \boldsymbol{S}_k \|_2} = \min_{\theta \in \Theta} \frac{1}{D_0 \times (T - t_0 + 1)} \sum_{k=1}^{D_0} \sum_{t=t_0}^{T} \frac{\| \boldsymbol{s}_k^t - \tilde{\boldsymbol{s}}_k^t \|_2}{\| \boldsymbol{s}_k^t \|_2} \quad (3)$$

where $\theta$ is a set of the network parameters and $\Theta$ is the parameter space.

## 4 METHODOLOGY

This section presents PAPM's architecture specifically tailored to conservation and constitutive relations. Then, a temporal-spatial stepping method (TSSM) is proposed, adapting to the unique

equation characteristics of various datasets. We conclude with a focus on our training approach, detailing the loss function and highlighting the strategy of iterative refinement rounds within a causal time-stepping training method.

## 4.1 PAPM OVERVIEW

Fig. 2 illustrates the PAPM, where the module comprises six distinct components. The input to the model consists of four parts, which are time $t$, coefficient $D$, initial state $\Phi_0$, and external source input $X_F$. The sequence of embedding this prior knowledge unfolds as follows: **1) Physics-informed Boundary Conditions.** Using the given boundary conditions, the physical quantity $\Phi_t$ is updated, yielding $\tilde{\Phi}_t$. A padding strategy is employed to integrate four different boundary conditions in four different directions into PAPM. Further details are provided in Appendix A.2. **2) Diffusive Flows (DF).** Using $\tilde{\Phi}_t$ and coefficients $\lambda$, we represent the directionless diffusive flow. The diffusion flow and its gradient are obtained as $J_D = -D \cdot \nabla\Phi$ and $\nabla J_D$ via a symmetric gradient operator, respectively. **3) Convective Flows (CF).** The pattern $v$ is derived from $\tilde{\Phi}_t$. Once $v$ is determined, its sign indicates the direction of the flows, enabling computation of $J_C = \Phi(x,t) \cdot v$ and $\nabla J_C$ through a directional gradient operator. **4) Internal Source Term (IST) & External Source Term (EST).** Generally, IST and EST present a complex interplay between physical quantities and external inputs $X_F$. Often, this part in real systems doesn't have a clear physics-based relation, prompting the use of NNs to capture this intricate relationship. **5) Time Stepping (TS).** From DF, CF, IST, and EST, the dynamic $\partial\Phi/\partial t$ are derived. Subsequently, the temporal operator is used to approximate the evolving state as $\Phi_{t+1} = \Phi_t + \int_t^{t+1} \frac{\partial\Phi}{\partial t} dt$.

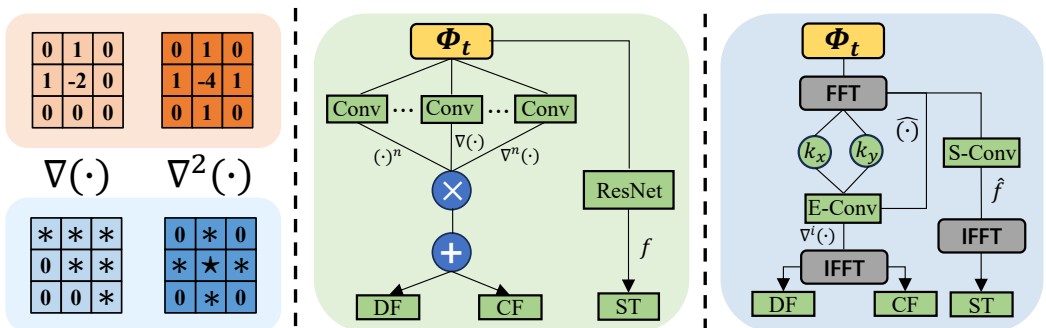

Figure 3: **Left:** Fixed and trainable convolutional kernels correspond to the matrices shown at the top and bottom, respectively, where $d = 3$ is used as an illustrative example. The bottom kernels approximate the unidirectional convection (upwind scheme) and directionless diffusion (central scheme), respectively. Symbols $*$ and $\star$ indicate trainable parameters corresponding to the upper triangular and symmetric matrices, respectively. **Mid:** Structure-preserved localized operator. **Right:** Structure-preserved spatial operator.

## 4.2 TEMPORAL-SPATIAL STEPPING METHOD (TSSM)

We categorize **TSSM** into three cases based on structures of process systems, where each of them decomposes spatial and temporal parts, *i.e.*, structure-preserved localized operator, spectral operator, and hybrid operator. Notably, despite adopting traditional time-stepping methods, such as Eular or RK4 schemes, a unique feature of our approach is the integration of the NN block. This element not only approximates unknown spatial components but also mitigates errors introduced by the time-stepping scheme. As a result, this dual functionality achieves enhanced stability, as evidenced by the consistent convergence of our training errors.

**Structure-preserved localized operator.** For systems with explicit structures, such as the Burgers and RD equations, typified by expressions like $-u\nabla u + \nabla^2 u$, convolutional kernels in the physical space are employed to capture system dynamics. Depending on our understanding of the system, we opt for either fixed or trainable kernels, illustrated in Fig. 3(Left). Specifically, the fixed variant is based on a pre-defined convolution kernel derived from difference schemes, and further details are provided in Appendix A.3.1. **On the other hand**, the trainable version tailors its design to key features of convection (either upper or lower triangular) and diffusion (symmetric). Once set, the localized operator is depicted in Fig. 3(Mid). By using these predefined or trainable convolution kernels, alongside partially known constitutive equations, we're able to represent nonlinear terms

in DF and CF. Any unknown source terms are then addressed through the ResNet block. For these explicit structures, the Euler scheme is introduced to establish the time-stepping update scheme.

**Structure-preserved spectral operator.** For systems with implicit structures, such as the Navier-Stokes Equation in vorticity form, represented like $-u\nabla w + \nabla^2 w$, we adopt a sequential process, as shown in Fig. 3(Right). Recognizing the implicit linkage between velocity and vorticity, $w$ is initially processed to extract the flow function, subsequently leading to the velocity derivation. The spectral space dimensions ($k_x$ and $k_y$) and spectral quantity (denoted as $\hat{\cdot}$), are obtained by leveraging the FFT. Associating $k_x$ and $k_y$ with $\hat{\cdot}$, differential operators like $\nabla^i(\cdot)$ are represented via E-Conv (*e.g.*, element-wise product), and then mapped back to the physical space using IFFT. This process can be further detailed in Appendix A.3.2. Leveraging partly known constitutive relations, simple computations such as addition and multiplication are performed to represent the nonlinear terms in DF and CF. Moreover, spectral convolutions (S-Conv) are introduced to learn unknown components within the spectral domain. For this complex process system with stiff challenges, such as fluid dynamics showcasing fast and slow time scale variations, the RK4 scheme is preferable to establish the time-stepping update scheme.

**Structure-preserved hybrid operator.** For systems with a hybrid structure, such as the Navier-Stokes Equation in general form (*e.g.*, $-u\nabla u + \nabla^2 u - \nabla p$), given the implicit interrelation between pressure $p$ and velocity $u$, a combination of the aforementioned method is employed. Explicit constituents, such as $u\nabla u$ and $\nabla^2 u$, are addressed through the localized operator. Meanwhile, implicit relations are resolved similarly by the spectral operator. For unknown components, either of the two operators can be engaged. We generally favor the localized operator as it allows for direct operations without necessitating transitions between different spaces.

### 4.3 Loss Function and Training strategy

Here, we detail the loss function and highlight the strategy of iterative refinement rounds within a causal time-stepping training method. As shown in Appendix A.4, Alg. 1 and Alg. 3 correspond to the structure-preserved localized operator and spectral operator, respectively. And the third one is a combination of the first two and will not be discussed here.

**Loss Function.** The mean $L_2$ relative error between the model's predictions and the true values spanning $N_t$ iterations is globally optimized, as illustrated in Eq. 4 (left side). Inspired by the insights from (Wang et al., 2022), the accuracy of previous outcomes directly dictates the precision of subsequent predictions within process systems. We propose a training strategy in forward optimization to address this inherent sequential linkage, named iterative refinement rounds.

$$\mathcal{L}_r(\boldsymbol{\theta}) = \frac{1}{N_t}\sum_{i=1}^{N_t}\mathcal{L}_r(t_i,\boldsymbol{\theta}) \Rightarrow \begin{cases} \mathcal{L}_r(\boldsymbol{\theta}) = \dfrac{1}{N_t}\sum_{i=1}^{N_t}w_i\mathcal{L}_r(t_i,\boldsymbol{\theta})\,, \\[2ex] w_i = \exp\left(-\alpha\sum_{k=1}^{i-1}\mathcal{L}_r(t_k,\boldsymbol{\theta})\right),\alpha\in\mathcal{R} \end{cases} \tag{4}$$

**Iterative Refinement Rounds.** By employing the causal time-stepping strategy (Wang et al., 2022), our loss function is a weighted residual loss with the inverse exponential of the residuals acting as weights $w_i$ (right side of Eq. 4). Moreover, the gradients of the $w_i$ term are detached to prevent gradient descent on the $\theta$ term within $w_i$. However, exponential decay has the potential to impact the model's subsequent results significantly. To mitigate this effect, we deploy three iterative training cycles, $[\alpha_1, \alpha_2, \alpha_3 = 0]$, adjusting only the causality parameter $\alpha$ to refine performance, where $\alpha_1 > \alpha_2 > \alpha_3 = 0$. Notably, when $\alpha = 0$, the weighted residual loss aligns with our initial loss.

## 5 Experiments

### 5.1 Experimental setup and evaluation protocol

**Datasets.** We employ five datasets spanning diverse domains, such as fluid dynamics and heat conduction. Detailed descriptions are available in Appendix A.5. By analyzing the TSSM scheme employed by PAPM, we've categorized these datasets accordingly, offering a structured understanding of their roles. **1. Localized Category. Burgers2d (Huang et al., 2023a)** is a 2D benchmark PDE with periodic BC given by the equation $\frac{\partial \boldsymbol{u}}{\partial t} = -\boldsymbol{u}\cdot\nabla\boldsymbol{u} + \nu\Delta\boldsymbol{u} + \boldsymbol{f}$. Here, $\boldsymbol{u} = (u_x, u_y)$ represents velocity. The aim is to predict subsequent frames of $\boldsymbol{u}$ under various initial conditions and viscosity $nu$

using the initial frames, while the forcing term $f$ remains unknown. **RD2d (Takamoto et al., 2022)** corresponds to a 2D F-N reaction-diffusion equation with no-flow Neumann BC, $\frac{\partial u}{\partial t} = \nu \Delta u + f$. Here, $u = (u, v)$ are the activator and inhibitor, respectively. The goal is to project subsequent frames under diverse initial conditions from the initial frames, with the source term as unknown. **2. Spectral Category. NS2d (Li et al., 2020)** is a dataset for incompressible fluid dynamics in vorticity form with periodic BC. The equation is $\frac{\partial w}{\partial t} = -u\nabla w + \nu\Delta w + f$, where $u$ is velocity, $w$ is vorticity, and $f$ is an unknown forcing term. The objective is to predict final frames from the initial frames of vorticity $w$ under varied initial conditions. **3. Hybrid Category. Lid2d** is a classical 2D dataset for incompressible lid-driven cavity flow with multiple BCs, $\frac{\partial u}{\partial t} = -u \cdot \nabla u + \nu\Delta u - \nabla p$. The goal is to predict subsequent frames $(u, v, p)$ based on initial ones at differing viscosity $\nu$, assuming only two flows are known. **NSM2d** is incompressible fluid dynamics with a magnetic field, described by $\frac{\partial u}{\partial t} = -u \cdot \nabla u + \nu\Delta u - \nabla p + F$. The target is to predict subsequent frames of $(u, v, p)$ using the initial frames, with the forcing term $F$ as an unknown.

**Baselines.** For a comprehensive evaluation, we compared our approach with eight SOTA baselines. **ConvLSTM** (Shi et al., 2015) is a classical time series modeling technique that captures dynamics via CNN and LSTM. **Dil-ResNet** (Stachenfeld et al., 2021) adopts the encoder-process-decoder process with dilated-ConvResNet for dynamic data through an autoregressive stepping manner. **time-FNO2D** (Li et al., 2020) and **MIONet** (Jin et al., 2022) are two typical neural operators in learning dynamics. **U-FNet** (Gupta & Brandstetter, 2022) and **CNO** (Raonić et al., 2023) are modified U-Net (Ronneberger et al., 2015) variants. **PeRCNN** (Rao et al., 2023) incorporates specific physical structures into a neural network, ideal for sparse data scenarios. **PPNN** (Liu et al., 2022) is a novel autoregressive framework preserving known PDEs using multi-resolution convolutional blocks.

**Metrics.** We use the mean $L_2$ relative error (abbreviated as $\epsilon$) as the evaluation metric. Suppose $\{S_i\}_{i=0}^{N_d}$ and $\{\tilde{S}_i\}_{i=0}^{N_d}$ are the ground-truth solution and the predicted solution respectively, where $N_d$ is the test dataset size. $S_i = (s_i^1, \cdots, s_i^{N_t})$, $\tilde{S}_i = (\tilde{s}_i^1, \cdots, \tilde{s}_i^{N_t})$ are sequence frames of length $N_t$. Here, $s_i^j, \tilde{s}_i^j \in \mathcal{R}^m$ denote the vector of $m$ physical quantities at this time slice. The $L_2$ error can be formulated as follows:

$$\epsilon = \frac{1}{N_d \times N_t \times m} \sum_{i=1}^{N_d} \sum_{j=1}^{N_t} \sum_{k=1}^{m} \frac{\|\tilde{s}_{i,k}^j - s_{i,k}^j\|_2}{\|s_{i,k}^j\|_2} \tag{5}$$

**Evaluation Protocol.** We conducted experiments in two settings: coefficient interpolation (referred to as **C Int.**) and coefficient extrapolation (referred to as **C Ext.**). In both settings, our objective is to **extrapolate time**. We consistently set the initial time step size for all datasets across various tasks as $t_0 = 5$. In the test set, $T_{end} = 100$, except for the NS2d. Specifically, for NS2d with viscosity values of $\nu = 1e-3$ and $1e-4$, $T_{end} = 50$, while for $\nu = 1e-5$, $T_{end} = 20$. The trajectory length in the training set that can be used as label data is given by $T_{end}/2 - t_0$. For **C Int.**, the data is uniformly shuffled and then split into training, validation, and testing datasets in a $[7 : 1 : 2]$ ratio. However, in the case of **C Ext.**, the data splitter is determined based on the order of coefficients, with equal proportions $[7 : 1 : 2]$. For example, the viscosity coefficients are divided from largest to smallest, and the coefficients with the lowest viscosity, representing the most challenging tasks, are selected as the test set. More information about the evaluation protocol, the hyper-parameters of baselines, and our methods can be further detailed in Appendix A.6.2.

## 5.2 MAIN RESULTS

**Performance Comparisons.** Tab. 1 and Tab. 2 present the primary experimental outcomes and the number of trainable parameters for each baseline across datasets, respectively. Our observations from the data are as follows:

**Firstly**, PAPM exhibits the most balanced trade-off between parameter count and performance among all methods evaluated, from explicit structures (Burgers2d, RD2d) to implicit (NS2d) and more complex hybrid structures (Lid2d, NSM2d). Notably, even though PAPM utilizes only 1% of the parameters employed by the prior leading method, PPNN, it still outperforms it by a large margin. In a nutshell, our model enhances the performance by an average of 6.4% over nine tasks, which affirms PAPM as a versatile and efficient framework suitable for diverse process systems.

**Secondly**, PAPM's structured treatment of system inputs and states leads to a remarkable 10.8% performance boost in three coefficient-extrapolation tasks. This highlights its superior generalization capability in out-of-sample scenarios. Unlike models like PPNN, which directly use system-specific

Table 1: Main results ($\epsilon$) across different datasets in time extrapolation task.

| Config | Burgers2d | | RD2d | NS2d | | | Lid2d | NSM2d | |
|---|---|---|---|---|---|---|---|---|---|
| | C Int. | C Ext. | C Int. | $\nu$=1e-3 | $\nu$=1e-4 | $\nu$=1e-5 | C Ext. | C Int. | C Ext. |
| ConvLSTM | 0.314 | 0.551 | 0.815 | 0.781 | 0.877 | 0.788 | 1.323 | 0.910 | 1.102 |
| Dil-ResNet | 0.071 | 0.136 | 0.021 | 0.152 | 0.511 | 0.199 | 0.261 | 0.288 | 0.314 |
| time-FNO2D | 0.173 | 0.233 | 0.333 | 0.118 | 0.100 | 0.033 | 0.265 | 0.341 | 0.443 |
| MIONet | 0.181 | 0.212 | 0.247 | 0.139 | 0.114 | 0.051 | 0.221 | 0.268 | 0.440 |
| U-FNet | 0.109 | 0.433 | 0.239 | 0.191 | 0.190 | 0.256 | 0.192 | 0.257 | 0.457 |
| CNO | 0.112 | 0.126 | 0.258 | 0.125 | 0.148 | 0.030 | 0.218 | 0.197 | 0.355 |
| PeRCNN | 0.212 | 0.282 | 0.773 | 0.571 | 0.591 | 0.275 | 0.534 | 0.493 | 0.493 |
| PPNN | 0.047 | 0.132 | 0.030 | 0.365 | 0.357 | 0.046 | 0.163 | 0.206 | 0.264 |
| **PAPM (Our)** | **0.039** | **0.101** | **0.018** | **0.110** | **0.097** | 0.034 | **0.160** | **0.189** | **0.245** |

Table 2: Comparison of the number of trainable parameters ($N_P$) across different datasets.

| Config | spatial/M | spectra/M | hybrid/M |
|---|---|---|---|
| ConvLSTM | 0.175 | 0.139 | 0.211 |
| Dil-ResNet | 0.150 | 0.148 | 0.152 |
| time-FNO2D | 0.464 | 0.463 | 0.464 |
| MIONet | 0.261 | 0.261 | 0.261 |
| U-FNet | 9.853 | 9.851 | 9.854 |
| CNO | 2.606 | 2.600 | 2.612 |
| PeRCNN | **0.001** | **0.001** | **0.001** |
| PPNN | 1.201 | 1.190 | 1.213 |
| **PAPM** | 0.014 | 0.034 | 0.035 |

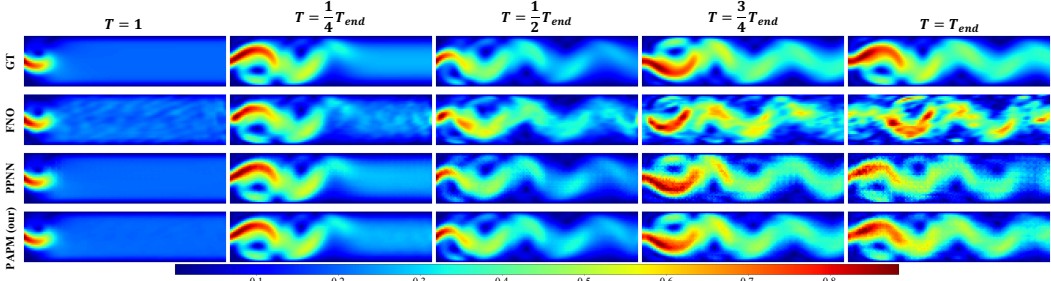

Figure 5: Predicted flow velocity ($\|\boldsymbol{u}\|_2$) snapshots by FNO, PPNN, and PAPM (Ours) vs. Ground Truth (GT) on NSM2d dataset in T Ext. task.

inputs, PAPM integrates coefficient data more intricately within conservation and constitutive relations, boosting its adaptability to varying coefficients.

**Thirdly**, data-driven methods are less effective than physics-aware methods like PPNN and our PAPM in time extrapolation tasks, where incorporating prior physical knowledge through structured network design enhances a model's generalization ability. Notably, PeRCNN uses $1 \times 1$ convolution to approximate nonlinear terms, but experimental results suggest limited performance. Further details are available in Appendix A.6.2.

**Visualization.** Fig. 4 showcases the stepwise relative error of PAPM during the extrapolation process in the test dataset, using Burgers2d's C Int. as a representative example. Compared to the two best-performing baselines, our model (depicted by the red line) exhibits superior performance throughout the extrapolation process, with the least error accumulation. Turning our attention to the more challenging NSM2d dataset, Fig. 5 presents the results across five extrapolation time slices. While FNO demonstrates commendable accuracy within the training domain $(T = \frac{1}{2}T_{end})$, its performance falters significantly outside of it $(\frac{1}{2}T_{end} < T \leq T_{end})$. On the other

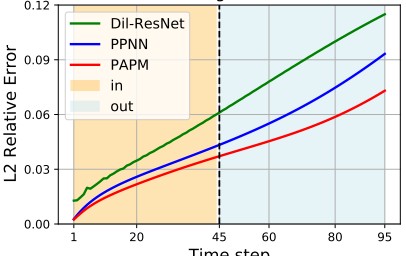

Figure 4: $L_2$ errors at each time step on Burgers2d, where **in** is the same as the training, but **out** is out-of-sample.

hand, physics-aware methods (PPNN), and PAPM in particular, consistently capture the evolving patterns with a greater degree of robustness. Notably, our method emerges as a leader in terms of precision. Additional visual results can be found in Appendix A.7.1.

**Performance and Computational Cost.** Tab. 3 showcases the comparison of primary results, computational cost (FLOPs), and the number of trainable parameters using the Lid2d dataset for illustration. Notably, PAPM strikes an optimal balance between performance and computational cost.

## 5.3 EFFICIENCY

**Training and Inference Cost**: Dataset generation for our work is notably resource-intensive, with inference costs ranging from $10^3 \sim 10^5$ s for public datasets and up to $10^6$ s for those we generated using COMSOL Multiphysics®. In stark contrast, both baselines and PAPM register inference times between $10^{-1} \sim 10$ s (detailed in Appendix A.7.2), achieving an improvement of 4 to 6 orders of magnitude. Notably, PAPM's time cost rivals or even surpasses baselines across different datasets. This indicates that introducing rigorous physical mechanisms, unlike traditional numerical

algorithms, doesn't necessarily bloat time costs. PAPM's efficiency remains competitive with other data-driven methods.

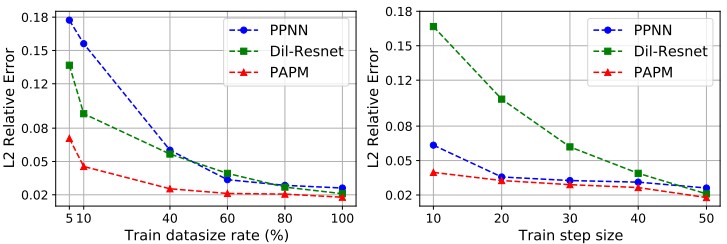

Figure 6: **Left**: Results with varying training data on the RD2d dataset. **Right**: Results with varying time step size on the RD2d dataset.

Table 3: Main results ($\epsilon$), FLOPs, and comparison of the number of trainable parameters ($N_P$) on Lid2d.

| Config | FLOPs | $N_p$ | $\epsilon$ |
|---|---|---|---|
| convLSTM | 327.54M | 211k | 1.323 |
| Dil-ResNet | 624.00M | 152k | 0.261 |
| timeFNO2d | 6.89M | 464k | 0.265 |
| MIONet | 6.89M | 261k | 0.221 |
| U-FNet | 559.89M | 9854k | 0.192 |
| CNO | 835.37M | 2612k | 0.218 |
| PeRCNN | 3.44M | 1k | 0.534 |
| PPNN | 348.56M | 1213k | 0.163 |
| **PAPM** | **1.22M** | 11k | **0.160** |

**Data Efficiency.** Owing to PAPM's structured design, data utilization is significantly enhanced. To evaluate data efficiency, we conducted tests using RD2d as a representative example, with Dil-Resnet and PPNN symbolizing pure data-driven and physics-aware methods. The results, displayed in Fig. 6, depict PAPM's efficiency concerning data volume and label data step size in training. **(1) Amount of Data**: With a fixed 20% reserved for the test set, the remaining 80% of the total data is allocated to the training set. We systematically varied the training data volume, ranging from initially utilizing only 5% of the training set and progressively increasing it to the entire 100%. PAPM's relative error distinctly outperforms other baselines, especially with limited data (5%). As depicted in Fig. 6(Left), PAPM's error consistently surpasses other methods, stabilizing below 2% as the training data volume increases. **(2) Time Step Size**: We varied the data step size from $1/10$ to half of the total, increasing in tenths. Results of Fig. 6(Right) reveal that PAPM can achieve long-range time extrapolation with minimal dynamic steps, consistently outshining other methods even with shorter training data step sizes.

## 5.4 ABLATION STUDIES

We selected the Burger2d dataset due to its representation of diffusion, convection, and force terms. Several configurations are defined to determine the effects of individual components. **no_DF** excludes diffusion, while **no_CF** omits convection. **no_Phy** retains only a structure with a residual connection, eliminating both diffusion and convection. The **no_BCs** setup removes explicit BCs embedding, **no_All** is purely data-driven, and **no_Iter** bypasses the Iterative Refinement Rounds training strategy.

Key findings include recognizing the crucial roles of diffusion and convection in dictating system dynamics. This was evident when the no_DF configuration showed that integrating the viscosity coefficient with the diffusion term was vital. Its absence led to significant errors, most notably in parameter extrapolation tasks. The necessity of boundary adherence to physical laws became clear with the no_BCs approach, as it notably reduced BC's relative errors. Lastly, the no_Iter setup accentuated the importance of the causal training strategy in the model's training process.

Table 4: Comparison of the main result ($\epsilon$) and the number of trainable parameters ($N_P$) on Burgers2d.

| Config | $N_P$ | T Ext. $\epsilon$ | T Ext. BC $\epsilon$ | C Ext. $\epsilon$ | C Ext. BC $\epsilon$ |
|---|---|---|---|---|---|
| no_DF | 13.90k | 0.067 | 0.051 | 0.207 | 0.067 |
| no_CF | 13.93k | 0.062 | 0.043 | 0.131 | 0.054 |
| no_Phy | 13.84k | 0.149 | 0.051 | 0.210 | 0.144 |
| no_BCs | 13.99k | 0.068 | 0.097 | 0.136 | 0.193 |
| no_All | 13.84k | 0.162 | 0.195 | 0.216 | 0.250 |
| no_Iter | 13.99k | 0.080 | 0.039 | 0.141 | 0.048 |
| **PAPM** | 13.99k | **0.039** | **0.037** | **0.101** | **0.043** |

## 6 CONCLUSION

In response to challenges posed by limited physical understanding and computational demands, we introduced PAPM, a novel physics-aware architecture. It uniquely incorporates partial prior mechanistic knowledge, including BCs, conservation, and constitutive relations specific to process systems. We extensively validated the efficacy of PAPM's structured design and its comprehensive spatio-temporal stepping modeling approach across five datasets and nine distinct out-of-sample tasks. Notably, PAPM achieved an average absolute performance improvement of 6.4%, requiring fewer FLOPs, and only 1% of the parameters employed by the prior leading method, PPNN. Through such analysis, the structural design and specialized spatio-temporal stepping method of PAPM exhibit the most balanced trade-off between accuracy and computational efficiency among all methods evaluated and an impressive out-of-sample generalization.

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

# A    APPENDIX

In this appendix, we first describe the process model further, showing in detail the starting point of our problem (A.1). Secondly, we perform a further theoretical presentation on the details of physical-informed boundary conditions (A.2, see in Fig. 2). Then, the technical details (A.3) of the proposed temporal and spatial stepping method (*i.e.*, TSSM) are further elaborated. Moreover, the algorithm display is detailed for the first two operators of TSSM (A.4). Subsequently, the five datasets are further described (A.5), where the generations of **Lid2d** and **NSM2d** are detailed by COMSOL multiphysics (A.5.2). The hyper-parameters settings of different baselines and PAPM are shown in detail (A.6), and some additional experimental results are shown. These are data visualization (A.7.1), training, and inference time cost-specific details (A.7.2). Then, we show another ability of PAPM, adaptive resolution (A.7.3). Here, we consider a classical multi-physics field coupling problem, low-temperature argon plasma discharge to show PAPM's excellent advantages when facing more complex process systems (A.8). Finally, the limitations and future work are discussed (A.9).

## A.1    PROCESS MODELS

Pivotal in engineering disciplines, process models serve to represent and predict the dynamics of diverse process systems, from entire plants to single equipment pieces. These models primarily rely on the interplay between conservation and constitutive equations, ensuring an accurate depiction of system dynamics. Conservation equations dictate the model's dynamics using partial differential equations that govern primary physical quantities like mass, energy, and momentum. On the other hand, constitutive equations relate potentials to extensive variables via algebraic equations, such as flows, temperatures, pressures, concentrations, and enthalpies, enriching the model's comprehensiveness. Additionally, the model's reliability is ensured by accounting for initial and boundary conditions, making these four components interdependently integral to the model's solid mathematical framework.

**Conservation equations.** The general form in differential representation is:

$$\frac{\partial \mathbf{\Phi}}{\partial t} = -\nabla \cdot (\boldsymbol{J}_C + \boldsymbol{J}_D) + \boldsymbol{q} + \boldsymbol{F} \tag{6}$$

where $\boldsymbol{J}_C = \mathbf{\Phi} \cdot \boldsymbol{v}$, $\boldsymbol{J}_D = -\boldsymbol{D} \cdot \nabla \mathbf{\Phi}$, and $\mathbf{\Phi}(\boldsymbol{x}, t)$ represents the state of the model, which is the object of our modeling, $\boldsymbol{x} \in \Omega$ and $t \in [0, T]$, $T \in \mathbb{R}_+$. $\boldsymbol{J}_C$ represents convective flows, and $\boldsymbol{J}_D$ represents diffusive or molecular flows. $\boldsymbol{v}$ describes the convective flow pattern into or out of the system volume. $\boldsymbol{D}$ represents the diffusion coefficient. $\boldsymbol{q}$, the internal source term, for example, the chemical reaction for component mass conservation, where species appear or are consumed due to reactions within the space of interest. Other internal source terms arise from energy dissipation, conversion, compressibility, or density changes. $\boldsymbol{F}$, the external source term, including gravitational, electrical, and magnetic fields as well as pressure fields.

**Constitutive equations.** For the internal source term, it usually depends on the state of the model and can be expressed as $\boldsymbol{q} = h_q(\mathbf{\Phi}, \boldsymbol{x}, t)$. The external source term is usually related to the external effects imposed and can be expressed as $\boldsymbol{F} = h_F(\boldsymbol{X}_F)$, where $\boldsymbol{X}_F$ is a vector of parameters imposed externally, which may include voltages, pressures, etc. For convective flows, the velocity $\boldsymbol{v}$ may be determined by the state of the model, which can be expressed as $\boldsymbol{v} = g(\mathbf{\Phi}, \boldsymbol{x}, t)$.

**Initial conditions (IC).** Every process or system evolves over time, but to predict or understand this evolution, we need a reference or a starting point. The initial conditions provide this starting point. For example, in the context of a reactor, initial conditions might describe the concentration of various reactants at $t = 0$. Mathematically, IC can be represented as $\mathbf{\Phi}(\boldsymbol{x}, 0) = \mathbf{\Phi}_0(\boldsymbol{x})$.

**Boundary conditions (BC).** Initial conditions set the foundation at $t = 0$, while boundary conditions inform how a system evolves and interacts with its environment, for instance, by specifying heat flux at a heat exchanger's boundary or flow rate at a reactor's inlet. These boundary conditions can be categorized as Dirichlet, prescribing specific values like temperature on the boundary; Neumann, defining derivatives or fluxes such as the heat flux; and Robin, which combines aspects of both Dirichlet and Neumann, encompassing parameters like both heat transfer rates and surface temperatures. Regardless of the type, they're mathematically expressed as $\mathbf{\Phi}(\boldsymbol{x}_b, t) = \boldsymbol{f}_b(t)$, where $\boldsymbol{x}_b \in \partial\Omega$.

## A.2  Embedding Boundary Conditions

This part covers the method of embedding four different boundary conditions (**Dirichlet**, **Neumann**, **Robin**, and **Periodic**) into neural networks via convolution padding. Let's consider a rectangular region in a $2D$ space, $\Omega = [0,a] \times [0,b]$, which can be discretized into an $M \times N$ grid, $\delta x = \frac{a}{M}$, $\delta y = \frac{b}{N}$. Each grid point can be represented as $X_{ij} = (x_i, y_j)$, where $i = 1,2,...,M$ and $j = 1,2,...,N$. Hence, we can transform the continuous space into a discrete grid of points.

**Boundary Conditions on the $X$-axis**. The direction vector is $\mathbf{n} = (1,0)^T$, which means the boundary conditions are the same for each $y$ value.

- **Dirichlet**: If the boundary condition is given as $\mathbf{\Phi}(X,t) = f(X,t), X \in \partial\Omega$, the discrete form would be $\mathbf{\Phi}_{Mj} = f_j$, and we can use a padding method in the convolution kernel $\mathbf{\Phi}_{Mj} = f_j$.

- **Neumann**: If the boundary condition is given as $\frac{\partial\mathbf{\Phi}(X,t)}{\partial\mathbf{n}} = f(X,t), X \in \partial\Omega$, the discrete form would be $\frac{\mathbf{\Phi}_{(M+1)j} - \mathbf{\Phi}_{(M-1)j}}{2\delta x} = f_j$ and we can use a padding method in the convolution kernel $\mathbf{\Phi}_{(M+1)j} = \mathbf{\Phi}_{(M-1)j} + (2 \times \delta x) \times f_j$.

- **Robin**: If the boundary condition is given as $\alpha\mathbf{\Phi}(X,t) + \beta\frac{\partial\mathbf{\Phi}(X,t)}{\partial\mathbf{n}} = f(X,t), X \in \partial\Omega$, the discrete form would be $\alpha\mathbf{\Phi}_{Mj} + \beta\frac{\mathbf{\Phi}_{(M+1)j} - \mathbf{\Phi}_{(M-1)j}}{2 \times \delta x} = f_j$. We can use a padding method in the convolution kernel $\mathbf{\Phi}_{(M+1)j} = \frac{2 \times \delta x}{\beta}(f_j - \alpha\mathbf{\Phi}_{Mj}) + \mathbf{\Phi}_{(M-1)j}$.

- **Periodic**: If the boundary condition is given as $\mathbf{\Phi}(X_1,t) = \mathbf{\Phi}(X_2,t), X_1 \in \partial\Omega_1, X_2 \in \partial\Omega_2$, where $\Omega_1$ denotes the left boundary and $\Omega_2$ the right boundary, the discrete form would be $\mathbf{\Phi}_{Mj} = \mathbf{\Phi}_{1j}$. We can use a padding method in the convolution kernel $\mathbf{\Phi}_{Mj} = \mathbf{\Phi}_{1j}, \mathbf{\Phi}_{(M+1)j} = \mathbf{\Phi}_{2j}$.

**Boundary Conditions on the $Y$-axis**. The direction vector is $\mathbf{n} = (0,1)^T$. The basic handling method is similar to the $x$-direction case but with the grid spacing replaced with $\delta y$, and the boundary conditions applied to the upper and lower boundaries, *i.e.*, $j = 1$ and $j = N$. The corresponding $y$-direction expressions can be derived by replacing $x$ with $y$ in the $x$-direction expressions and swapping $i$ with $j$.

**Arbitrary Direction Boundary Conditions in the Rectangular Area.** The direction vector $\mathbf{n} = (cos(\theta), sin(\theta))^T$. Both $x$ and $y$ directions need to be considered, resulting in the following expressions for each of the four boundary conditions:

- **Dirichlet**:Given the condition $\mathbf{\Phi}(X,t) = f(X,t)$, its discrete form remains $\mathbf{\Phi}_{ij} = f_{ij}$. The corresponding padding method in the convolution kernel is $\mathbf{\Phi}_{ij} = f_{ij}$.

- **Neumann**: For the boundary condition $\frac{\partial\mathbf{\Phi}(X,t)}{\partial\mathbf{n}} = f(X,t)$, the discrete form can be represented as $cos(\theta)\frac{\mathbf{\Phi}_{(i+1)j} - \mathbf{\Phi}_{(i-1)j}}{2\delta x} + sin(\theta)\frac{\mathbf{\Phi}_{i(j+1)} - \mathbf{\Phi}_{i(j-1)}}{2\delta y} = f_{ij}$. The corresponding padding method in the convolution kernel can be written as $\mathbf{\Phi}_{(i+1)j} = \mathbf{\Phi}_{(i-1)j} + 2cos(\theta)\delta x f_{ij}$ and $\mathbf{\Phi}_{i(j+1)} = \mathbf{\Phi}_{i(j-1)} + 2sin(\theta)\delta y f_{ij}$.

- **Robin**: Given the condition $\alpha\mathbf{\Phi}(X,t) + \beta\frac{\partial\mathbf{\Phi}(X,t)}{\partial\mathbf{n}} = f(X,t)$, the discrete form becomes $\alpha\mathbf{\Phi}_{ij} + \beta cos(\theta)\frac{\mathbf{\Phi}_{(i+1)j} - \mathbf{\Phi}_{(i-1)j}}{2\delta x} + \beta sin(\theta)\frac{\mathbf{\Phi}_{i(j+1)} - \mathbf{\Phi}_{i(j-1)}}{2\delta y} = f_{ij}$. The corresponding padding method in the convolution kernel is $\mathbf{\Phi}_{(i+1)j} = \frac{1}{\beta cos(\theta)}[f_{ij} - \alpha\mathbf{\Phi}_{ij}] \times 2\delta x + \mathbf{\Phi}_{(i-1)j}$ and $\mathbf{\Phi}_{i(j+1)} = \frac{1}{\beta sin(\theta)}[f_{ij} - \alpha\mathbf{\Phi}_{ij}] \times 2\delta y + \mathbf{\Phi}_{i(j-1)}$.

- **Periodic**: For the condition $\mathbf{\Phi}(X_1,t) = \mathbf{\Phi}(X_2,t)$, the discrete form is $\mathbf{\Phi}_{Mj} = \mathbf{\Phi}_{1j}$ and $\mathbf{\Phi}_{Ni} = \mathbf{\Phi}_{1i}$. The corresponding padding method in the convolution kernel is $\mathbf{\Phi}_{Mj} = \mathbf{\Phi}_{1j}, \mathbf{\Phi}_{(M+1)j} = \mathbf{\Phi}_{2j}$ and $\mathbf{\Phi}_{Ni} = \mathbf{\Phi}_{1i}, \mathbf{\Phi}_{i(N+1)} = \mathbf{\Phi}_{i2}$.

**Directionless Boundary Conditions in the Rectangular Area.** The following strategies are employed for handling the Neumann and Robin boundary conditions:

- For $\frac{\partial \Phi}{\partial X} = f(X, t)$, the discrete form is $\frac{\Phi_{(i+1)j} - \Phi_{(i-1)j}}{2\delta x} = f_{ij}$ and $\frac{\Phi_{i(j+1)} - \Phi_{i(j-1)}}{2\delta y} = f_{ij}$. We can use a padding method in the convolution kernel where $\Phi_{(i+1)j} = \Phi_{(i-1)j} + 2\delta x f_{ij}$ and $\Phi_{i(j+1)} = \Phi_{i(j-1)} + 2\delta y f_{ij}$.

- For $\alpha\Phi(X, t) + \beta\frac{\partial \Phi(X, t)}{\partial X} = f(X, t)$, the discrete form is $\alpha\Phi_{ij} + \beta\frac{\Phi_{(i+1)j} - \Phi_{(i-1)j}}{2\delta x} = f_{ij}$ and $\alpha\Phi_{ij} + \beta\frac{\Phi_{i(j+1)} - \Phi_{i(j-1)}}{2\delta y} = f_{ij}$. We can use a padding method in the convolution kernel where $\Phi_{(i+1)j} = \frac{1}{\beta}[f_{ij} - \alpha\Phi_{ij}] \times 2\delta x + \Phi_{(i-1)j}$ and $\Phi_{i(j+1)} = \frac{1}{\beta}[f_{ij} - \alpha\Phi_{ij}] \times 2\delta y + \Phi_{i(j-1)}$.

## A.3 TEMPORAL-SPATIAL STEPPING METHOD

### A.3.1 STRUCTURE-PRESERVED LOCALIZED OPERATOR

**Fixed convolution operations.** The differential operator can be approximated via convolution operations. For a one-dimensional function $u(x)$, we could use a convolution kernel of the form:

$$K = \frac{1}{2\Delta x}[-1, 0, 1] \tag{7}$$

where $\Delta x$ represents the step size. This convolution operation, corresponding to this kernel, can approximate the first-order central difference operator as follows:

$$u'(x) \approx \frac{u(x + \Delta x) - u(x - \Delta x)}{2\Delta x} \approx u(x) \circledast K, \tag{8}$$

with $\circledast$ denoting the convolution operation. For a two-dimensional function, it can be decomposed into a convolution of two one-dimensional functions. Assuming $u(x, y)$ is a two-dimensional function, the kernel could be formed as:

$$K = \frac{1}{h^2}\begin{bmatrix} 0 & 1 & 0 \\ 1 & -4 & 1 \\ 0 & 1 & 0 \end{bmatrix} = \frac{1}{h^2}\begin{bmatrix} 0 & 0 & 0 \\ 1 & -2 & 1 \\ 0 & 0 & 0 \end{bmatrix} + \frac{1}{h^2}\begin{bmatrix} 0 & 1 & 0 \\ 0 & -2 & 0 \\ 0 & 1 & 0 \end{bmatrix}, \tag{9}$$

where $h = \Delta x = \Delta y$ signifies the step size. The convolution operation corresponding to this kernel can approximate the second-order central difference operator, which is:

$$
\begin{aligned}
\nabla^2 u(x, y) &= \frac{\partial^2 u(x, y)}{\partial x^2} + \frac{\partial^2 u(x, y)}{\partial y^2} \\
&\approx \frac{u(x + h, y) - 2u(x, y) + u(x - h, y)}{h^2} + \frac{u(x, y + h) - 2u(x, y) + u(x, y - h)}{h^2} \\
&= \frac{u(x + h, y) + u(x, y + h) - 4u(x, y) + u(x - h, y) + u(x, y - h)}{h^2} \\
&= u(x, y) \circledast K.
\end{aligned}
\tag{10}
$$

Analogously, different convolution kernels can approximate other orders' differential operators. Utilizing convolution operations to approximate differential operators can boost computational efficiency. Nevertheless, careful consideration is needed when choosing a convolution kernel, as different kernels can influence the stability and accuracy of the numerical solution.

**Selection of FD kernels.** FD kernels are used to approximate derivative terms in PDEs, which directly affect the computational efficiency and the reconstruction accuracy. Therefore, it is crucial to choose appropriate FD kernels for discretized-based learning frameworks. For spatio-temporal systems, we need to consider both temporal and spatial derivatives. In specific, the second-order central difference is utilized for calculating temporal derivatives, *i.e.*,

$$\frac{\partial u}{\partial t} = \frac{-u(t - \delta t, \xi, \eta) + u(t + \delta t, \xi, \eta)}{2\delta t} + \mathcal{O}\left((\delta t)^2\right), \tag{11}$$

where $\{\xi, \eta\}$ represent the spatial locations and $\delta t$ is time spacing. In the network implementation, it can be organized as a convolutional kernel $K_t$,

$$K_t = [-1, 0, 1] \times \frac{1}{2\delta t}.$$

Likewise, we also apply the central difference to calculate the spatial derivatives for internal nodes and use forward/backward differences for boundary nodes. For instance, in this paper, the fourth-order central difference is utilized to approximate the first and second spatial derivatives. The FD kernels for 2D cases with the shape of $5 \times 5$ are given by

$$K_{s,1} = \frac{1}{12(\delta x)} \begin{bmatrix} 0 & 0 & 0 & 0 & 0 \\ 0 & 0 & 0 & 0 & 0 \\ 1 & -8 & 0 & 8 & -1 \\ 0 & 0 & 0 & 0 & 0 \\ 0 & 0 & 0 & 0 & 0 \end{bmatrix}, K_{s,2} = \frac{1}{12(\delta x)^2} \begin{bmatrix} 0 & 0 & -1 & 0 & 0 \\ 0 & 0 & 16 & 0 & 0 \\ -1 & 16 & -60 & 16 & -1 \\ 0 & 0 & 16 & 0 & 0 \\ 0 & 0 & -1 & 0 & 0 \end{bmatrix},$$

(12)

where $\delta x$ denotes the grid size of HR variables; $K_{s,1}$ and $K_{s,2}$ are FD kernels for the first and second derivatives, respectively. In addition, we conduct a parametric study on the selection of FD kernels, including the second-order $(3 \times 3)$, the fourth-order $(5 \times 5)$, and the sixth-order $(7 \times 7)$ central difference strategies.

### A.3.2 STRUCTURE-PRESERVED SPECTRAL OPERATOR

In Fourier space, the simplification of problem-solving involves converting differential operations and wave number multiplications. In 2D space, the discrete values of the set function $\Phi(x, y)$ in real space, denoted as $\Phi_{ij}$, correspond to $\hat{\Phi}_{mn}$ in Fourier space, with $k_x$ and $k_y$ representing the respective wave numbers. Differential operators are transformed as follows. (1) The first-order differential operator, $\frac{\partial \Phi}{\partial x}$, becomes $ik_x \hat{\Phi}_{mn}$ in the $x$ direction and $ik_y \hat{\Phi}_{mn}$ in the $y$ direction. (2) The second-order differential operator, $\frac{\partial^2 \Phi}{\partial x^2}$, is represented as $-k_x^2 \hat{\Phi}_{mn}$ for the $x$ direction and $-k_y^2 \hat{\Phi}_{mn}$ for the $y$ direction. (3) The Laplacian operator $\nabla^2 \Phi$ transforms into $(-k_x^2 - k_y^2)\hat{\Phi}_{mn}$ in Fourier space.

Moreover, for a 2D flow field, the relationship between the flow function $\psi$ and the velocity fields $(u, v)$ can be expressed by the following partial differential equation, $u = \frac{\partial \psi}{\partial y}$ and $v = -\frac{\partial \psi}{\partial x}$, where we can use $k_x$ and $k_y$ to obtain differential results. Moreover, here **S-Conv** is from FNO (Li et al., 2020) (the module named as "SpectralConv2d_fast"). This "SpectralConv2d_fast" class is a neural network module that performs a 2D spectral convolution by applying an FFT, a learned linear transformation in the Fourier domain, and an IFFT.

### A.4 ALGORITHM DISPLAY

Here, we detail the loss function and highlight the strategy of iterative refinement rounds within a causal time-stepping training method. As shown in Alg. 1 and Alg. 3, the first two operators are detailed, and the third one is a combination of the first two and will not be discussed here.

---

**Algorithm 1:** Structure-preserved localized operator with iterative refinement rounds.

---

**Initialization:** Fixed or pre-defined convolutional kernels $K$ (with parameters $\theta$), as shown in Fig. 3 (Left); Initialize other network parameters $\theta \in \Theta$;

**Input:** A set of inputs $\boldsymbol{a}_k$ for $1 \le k \le D_0$, initial step size $t_0$, and temporal trajectory length $T'$, time interval $\Delta t$, iterative refinement rounds parameters $[\alpha_1, \alpha_2, \alpha_3]$;

**Output:** The mapping $\mathcal{G}_\theta$, where $\tilde{\boldsymbol{S}}_k \leftarrow \tilde{\mathcal{G}}_\theta(\boldsymbol{a}_k)$;

**Require:** Parameterized neural network $\tilde{\mathcal{G}}_\theta$, and iterative refinement rounds strategy ;

**for** $\alpha$ in $[\alpha_1, \alpha_2, \alpha_3]$ **do**

    **for** $k = 1$ *to* $D_0$ **do**

        $\boldsymbol{a}_k \leftarrow [\boldsymbol{s}_k^0, \boldsymbol{s}_k^1, \ldots, \boldsymbol{s}_k^{t_0-1}]$ ;

        **for** $t = t_0$ *to* $T'$ **do**

            $(\cdot)^n, \nabla^n(\cdot), n = 0, 1, \ldots \leftarrow \boldsymbol{a}_k \circledast K$;

            DF, CF, IST, EST $\leftarrow (\cdot)^n, \nabla^n(\cdot), \boldsymbol{a}_k$;

            TS $\leftarrow$ DF+CF+IST+EST;

            Next states $\boldsymbol{s}_k^{t+1} \leftarrow \boldsymbol{s}_k^t + \text{TS} \times \Delta t$;

            Update input $\boldsymbol{a}_k \leftarrow \boldsymbol{s}_k^{t+1}$;

        **end**

        Subsequent trajectory $\tilde{\boldsymbol{S}}_k \leftarrow \tilde{\mathcal{G}}_\theta(\boldsymbol{a}_k)$ ;

        Loss $\mathcal{L}_r(\boldsymbol{\theta}) \leftarrow$ Eq. 4 ;

        Detach gradients of $w_i$ to prevent gradient descent on $\theta$ within $w_i$ ;

        Update weights $\theta$ by minimizing the loss $\mathcal{L}_r(\boldsymbol{\theta})$ ;

    **end**

    Adjust causality parameter $\alpha$ for the next refinement round;

**end**

---

**Algorithm 2:** Update State by Structure-Preserved Spectral Operator

---

**Input:** $\boldsymbol{a}_k$ and time interval $\Delta t$

**Output:** Next states $\boldsymbol{s}_k^{t+1}$

1. $k_x, k_y, \hat{\cdot} \leftarrow \text{FFT}(\boldsymbol{a}_k)$;
2. $\hat{(\cdot)}^n, \hat{\nabla}^n(\cdot), n = 0, 1, \ldots \leftarrow \text{E-Conv}(k_x, k_y, \hat{\cdot})$;
3. DF, CF $\leftarrow (\cdot)^n, \nabla^n(\cdot), \boldsymbol{a}_k$;
4. IST, EST $\leftarrow \text{S-conv}(\cdot)$;
5. TS $\leftarrow$ DF $+$ CF $+$ IST $+$ EST;
6. $\Delta \boldsymbol{s}_k^{t+1} \leftarrow \boldsymbol{s}_k^t + \text{TS} \times \Delta t$;

---

---

**Algorithm 3:** Structure-Preserved Spectral Operator with Iterative Refinement Rounds

---

**Initialization:** Initialize E-Conv ($1 \times 1$ conv) and S-conv (spectral convolutions) with parameters $\theta$ as shown in Fig. 3 (Right);

**Input:** A set of inputs $\boldsymbol{a}_k$ for $1 \leq k \leq D_0$, initial step size $t_0$, temporal trajectory length $T'$, time interval $\Delta t$, iterative refinement rounds parameters $[\alpha_1, \alpha_2, \alpha_3]$

**Output:** The mapping $\mathcal{G}_\theta$ where $\tilde{\boldsymbol{S}}_k \leftarrow \tilde{\mathcal{G}}_\theta(\boldsymbol{a}_k)$

**Require:** Parameterized neural network $\tilde{\mathcal{G}}_\theta$ and iterative refinement rounds strategy

**foreach** $\alpha$ in $[\alpha_1, \alpha_2, \alpha_3]$ **do**
    **for** $k = 1$ **to** $D_0$ **do**
        $\boldsymbol{a}_k \leftarrow [\boldsymbol{s}_k^0, \boldsymbol{s}_k^1, \ldots, \boldsymbol{s}_k^{t_0-1}]$;
        **for** $t = t_0$ **to** $T'$ **do**
            $r_1 \leftarrow \text{Alg2}(\boldsymbol{a}_k, 0)$;
            $\boldsymbol{a}_k \leftarrow \boldsymbol{s}_k^t + r_1 \times \frac{\Delta t}{2}$;
            $r_2 \leftarrow \text{Alg2}(\boldsymbol{a}_k, \frac{\Delta t}{2})$;
            $\boldsymbol{a}_k \leftarrow \boldsymbol{s}_k^t + r_2 \times \frac{\Delta t}{2}$;
            $r_3 \leftarrow \text{Alg2}(\boldsymbol{a}_k, \frac{\Delta t}{2})$;
            $\boldsymbol{a}_k \leftarrow \boldsymbol{s}_k^t + r_3 \times \Delta t$;
            $r_4 \leftarrow \text{Alg2}(\boldsymbol{a}_k, \Delta t)$;
            $\boldsymbol{s}_k^{t+1} \leftarrow \boldsymbol{s}_k^t + \frac{\Delta t}{6}(r_1 + 2r_2 + 2r_3 + r_4)$;
            Update input: $\boldsymbol{a}_k \leftarrow \boldsymbol{s}_k^{t+1}$;
        **end**
        $\tilde{\boldsymbol{S}}_k \leftarrow \tilde{\mathcal{G}}_\theta(\boldsymbol{a}_k)$ ;
        Compute loss $\mathcal{L}_r(\boldsymbol{\theta})$ according to Eq. 4 ;
        Detach gradients of $w_i$ to inhibit gradient descent on $\theta$ within $w_i$ ;
        Update weights $\theta$ by minimizing the loss $\mathcal{L}_r(\boldsymbol{\theta})$ ;
    **end**
    Adjust the causality parameter $\alpha$ for the next refinement round;
**end**

---

A.5 DATASETS

We employ five datasets spanning diverse domains, such as fluid dynamics and heat conduction. Based on the TSSM scheme employed by PAPM, we categorize the aforementioned five datasets into three types: **Burgers2d** and **RD2d** fall under the **localized** category, **NS2d** is classified as **spectral**, while **Lid2d** and **NSM2d** are designated as **hybrid**. The generations of Lid2d and NSM2d are detailed via COMSOL multiphysics in A.5.2. We are particularly keen to make **Lid2d** and **NSM2d** publicly available, anticipating various research endeavors on these datasets by the community.

A.5.1 FIVE DATASETS

**Burgers2d (Huang et al., 2023a).** The 2D Burgers equation is one of the fundamental nonlinear partial differential equations. Its formulation is given by:

$$\begin{cases} \dfrac{\partial \boldsymbol{u}}{\partial t} = -\boldsymbol{u} \cdot \nabla \boldsymbol{u} + v\Delta \boldsymbol{u} + \boldsymbol{f}, \\ \boldsymbol{u}|_{t=0} = \boldsymbol{u}_0(x,y) \end{cases} \tag{13}$$

where $\boldsymbol{u} = (u(x,y,t), v(x,y,t))$ represents the velocity field, and the spatial domain is $\Omega = [0, 2\pi]^2$ with periodic boundary conditions. The viscosity coefficient $v$ varies within the range $v \in [0.001, 0.1]$. The forcing term is defined as:

$$f(x, y, \boldsymbol{u}) = (\sin(v)\cos(5x + 5y), \sin(u)\cos(5x - 5y))^\top. \tag{14}$$

The initial condition, denoted as $\boldsymbol{u}_0(x,y)$, is drawn from a Gaussian random field characterized by a variance of $25(-\Delta + 25I)^{-3}$. Subsequently, it is linearly normalized to fall within the range of $[0.1, 1.1]$. A total of $N = 500$ samples are generated, each spanning $M = 3200$ time steps with a step size of $\delta t = \frac{0.01}{32}$. For the generation of high-precision numerical solutions, a high-resolution traditional numerical solver is employed. This solver utilizes the value of $\delta t$ and operates on a finely discretized $256 \times 256$ grid. The resulting high-precision solutions are stored at intervals of every 32 time step, resulting in 100 time slices. Subsequently, these solutions are downsampled to a coarser $64 \times 64$ grid.

**RD2d (Takamoto et al., 2022).** Considering the 2D diffusion-reaction equation, the conservation of the activator $u$ and inhibitor $v$ can be represented as:

$$\begin{cases} \dfrac{\partial u}{\partial t} = -\nabla J_u + R_u, \dfrac{\partial v}{\partial t} = -\nabla J_v + R_v \\ J_u = -D_u\nabla u, \quad J_v = -D_v\nabla v \end{cases} \tag{15}$$

Where $J_u$ and $J_v$ are the flux terms for the activator and inhibitor, respectively. These represent the diffusive or molecular flows for each component. The reaction functions $R_u$ and $R_v$ for the activator and inhibitor, respectively, are defined by the Fitzhugh-Nagumo (FN) equation, written as $R_u = u - u^3 - k - v$ and $R_v = u - v$, where $k = 5 \times 10^{-3}$ and the diffusion coefficients for the activator and inhibitor are $D_u = 1 \times 10^{-3}$ and $D_v = 5 \times 10^{-3}$, respectively. The initial condition is characterized by a standard normal random noise, with $u(0, x, y) \sim \mathcal{N}(0, 1.0)$ for $x \in (-1, 1)$ and $y \in (-1, 1)$. The boundary conditions are defined as no-flow Neumann boundary conditions. This entails that the partial derivatives satisfy the conditions: $D_u\partial_x u = 0$, $D_v\partial_x v = 0$, $D_u\partial_y u = 0$, and $D_v\partial_y v = 0$, all applicable for the domain $x, y \in (-1, 1)^2$. This dataset [1] is transformed into a coarser grid with dimensions of $64 \times 64$ while keeping the time step consistently constant.

**NS2d (Li et al., 2020).** We refer to **FNO** as the source for our exploration of the two-dimensional incompressible Navier-Stokes equation in vorticity form. This equation is defined on the unit torus and is outlined as follows:

$$\begin{cases} \partial_t w(x,t) + u(x,t) \cdot \nabla w(x,t) = \nu\Delta w(x,t) + f, \\ \nabla \cdot u(x,t) = 0, \\ w(x,0) = w_0(x), \end{cases} \tag{16}$$

where $x \in (0,1)^2$, $t \in (0, T]$, and $u$ represents the velocity field, $w = \nabla \times u$ denotes the vorticity, $w_0$ stands for the initial vorticity distribution, $\nu \in \mathbb{R}_+$ signifies the viscosity coefficient and $f$ denotes

---

[1]This dataset can be downloaded at https://github.com/pdebench/PDEBench

the forcing function. In this work, the viscosity coefficient is set to $\nu = 1 \times 10^{-3}, 1 \times 10^{-4}, 1 \times 10^{-5}$. It's worth noting that, for the purpose of maintaining a consistent evaluation framework, the resolution is standardized at $64 \times 64$ for both training and testing phases, given that the baseline methods are not inherently resolution-invariant.

**Lid2d.** A constant velocity across the top of the cavity creates a circulating flow inside. To simulate this, a constant velocity boundary condition is applied to the lid while the other three walls obey the no-slip condition. Different Reynolds numbers yield different results, so in this article, $Re \in [100, 1500]$ are applied. At high Reynolds numbers, secondary circulation zones are expected to form in the corners of the cavity. The system of differential equations (N-S equations) consists of two equations for the velocity components $\boldsymbol{u} = (u(x, y, t), v(x, y, t))$, and one equation for pressure $(p(x, y, t))$:

$$\begin{cases} \dfrac{\partial \boldsymbol{u}}{\partial t} = -\boldsymbol{u} \cdot \nabla \boldsymbol{u} + \dfrac{1}{Re} \Delta \boldsymbol{u} - \nabla p, \\ \nabla \cdot \boldsymbol{u} = 0 \end{cases} \tag{17}$$

where $(x, y) \in (0, 1)^2$. The initial condition is $(u, v, p) = \boldsymbol{0}$ everywhere. And the boundary conditions are: $u = 1$ at $y = 1$ (the lid), $(u, v) = \boldsymbol{0}$ on the other boundaries, $\partial p / \partial y = 0$ at y=0,p=0 at $y = 1$, and $\partial p / \partial x = 0$ at $x = 0, 1$. The data generation for the Lid2d is processed by COMSOL Multiphysics®, and a total of $N = 500$ samples are generated, each spanning $M = 1000$ time steps with a step size of $\delta t = \frac{0.1}{10}$. Every 10 steps, we save the data, resulting in 100 time slices. This solver utilizes the value of $\delta t$ and operates on a finely discretized $128 \times 128$ grid. Subsequently, these solutions are downsampled to a coarser $64 \times 64$ grid.

**NSM2d.** Consider the Navier-Stokes equations with an additional magnetic field:

$$\begin{cases} \dfrac{\partial \boldsymbol{u}}{\partial t} + \boldsymbol{u} \cdot \nabla \boldsymbol{u} = -\nabla p + \nu \nabla^2 \boldsymbol{u} + \boldsymbol{F}, \quad t \in [0, T], \\ \nabla \cdot \boldsymbol{u} = 0, \end{cases} \tag{18}$$

where $(x, y) \in [0, 4] \times [0, 1]$, $\boldsymbol{u} = [u(x, y, t), v(x, y, t)] \in \mathbb{R}^2$ is the velocity vector, $p(x, y, t) \in \mathbb{R}$ is the pressure, $\nu = 1/Re$ represents the kinematic viscosity (with $Re$ as the Reynolds number), and $\boldsymbol{F} = [F_x, F_y]$ is an external source term induced by the magnetic field. The components of $\boldsymbol{F}$ are defined as follows:

$$\begin{cases} F_x = mH \dfrac{\partial H}{\partial x}, \quad F_y = mH \dfrac{\partial H}{\partial y} \\ H(x, y) = \exp\left[-8\left((x - L/2)^2 + (y - W/2)^2\right)\right] \end{cases} \tag{19}$$

where $L = 4$, $W = 1$, $m = 0.16$ is the magnetization, and $H$ is a time-invariant magnetic intensity. The simulation is conducted on a 2D rectangular domain $\{x, y\} \in [0, 4] \times [0, 1]$ with the following boundary conditions: the inflow boundary $(x = 0)$ is prescribed with a velocity distribution $\boldsymbol{u}(0, y, t)$, where $y_0$ represents the vertical position of the inlet jet center:

$$\boldsymbol{u}(0, y, t) = \begin{bmatrix} u(0, y, t) \\ v(0, y, t) \end{bmatrix} = \begin{bmatrix} \exp\left(-50\left(y - y_0\right)^2\right) \\ \sin(t) \cdot \exp\left(-50\left(y - y_0\right)^2\right) \end{bmatrix} \tag{20}$$

The outflow boundary $(x = 4)$ is set with a reference pressure $p(4, y, t) = 0$. The no-slip boundary condition is applied at the top and bottom walls $(y = 0, 1)$. The Reynolds number is dimensionless and ranges from 100 to 1500. The inlet jet position $y_0$ is varied within the domain $0.4 \leq y_0 \leq 0.6$. The data generation for the NSM2d is processed by COMSOL Multiphysics®, and a total of $N = 500$ samples are generated, each spanning $M = 1000$ time steps with a step size of $\delta t = \frac{0.2}{10}$. Every 10 steps, we save the data, resulting in 100 time slices. This solver utilizes the value of $\delta t$ and operates on a finely discretized $256 \times 64$ grid. Subsequently, these solutions are downsampled to a coarser $128 \times 32$ grid.

### A.5.2 DETAILED DATA GENERATION PROCESS

Our research employed COMSOL multiphysics software [2] for fluid dynamics simulation in a lid-driven cavity and a magnetic stirring scenario. The simulation parameters are outlined in the main

---

[2]https://www.comsol.com/

text, utilizing grids of $128 \times 128$ and $256 \times 64$ for each case, respectively. The Time-Dependent Module, with specific time steps, was used for execution. The simulations required substantial computational resources, solving for 49,152 and 16,130 internal degrees of freedom (DOFs) in each scenario. To generate a comprehensive dataset, we varied simulation parameters, running 500 simulations for each scenario with different Reynolds numbers and, in the magnetic stirring case, the $y_0$ value. This data was stored in h5 format.

The computational intensity was significant: a single run in the lid-driven scenario took 91 seconds on average, while the magnetic stirring case took 226 seconds. The total computation time was approximately $10^6$ seconds for all 500 cases, highlighting the time-consuming nature of such simulations. The intricacy of multi-physics coupling and the extensive computational demand in these simulations point towards the necessity of more efficient methods. This situation underscores the potential of neural networks in accelerating simulation processes. By leveraging neural networks, we aim to reduce the computational time significantly, addressing the inherent slowness of detailed simulations like those in our study. This approach could transform the feasibility and scalability of complex simulations in various scientific and engineering domains.

### A.6 HYPER-PARAMETERS AND DETAILS FOR MODELS

#### A.6.1 EXPERIMENTAL SETUP

We train all models with AdamW (Loshchilov & Hutter, 2017) optimizer with the exponential decaying strategy, and epochs are set as 500. The causality parameter $\alpha_1 = 0.1$ and $\alpha_0 = 0.001$. The initial learning rate is $1e\text{-}3$, and the ReduceLRonPlateau schedule is utilized with a patience of 20 epochs and a decay factor of $0.8$. For a fair comparison, the batch size is identical across all methods for the same task, and all experiments are run on $1 \sim 3$ NVIDIA Tesla P100 GPUs.

To account for potential variability due to the partitioning process, each experiment is performed three times, and the final result is derived as the average of these three independent runs. Except for the predefined parameters, the parameters of all models are initialized by Xavier (Glorot & Bengio, 2010), setting the scaling ratio $c = 0.02$.

#### A.6.2 HYPER-PARAMETERS

**PAPM.** In this work, PAPM designed three different temporal-spatial modeling methods according to the characteristics of five different data sets.

- **Localized Operator.** For Burgers2d and RD2d, the localized operator is selected in spatial, while the Euler scheme is selected in temporal. Burgers2d uses the predefined fixed convolution kernel as the convolution kernel parameter of diffusive and convective flows, while a 4-layer convolution layer characterizes the source term. Its channel is set to 16, and the GELU activation function is used. In RD2d, trainable convolutional kernels are used as the convolution kernel parameter of diffusive flows, and the kernel is set to 5. For the source term, like burgers2d, a four-layer convolutional layer with channel 16 and GELU is used to characterize the source term.

- **Spectral Operator.** In NS2d, the spectral operator is selected in spatial, while the RK-4 scheme is selected in temporal. After FFT, $k_x$, $k_y$ and $\hat{w}$ are input into a $1 \times 1$ conv for dot product, which is used to solve the partial derivatives of vorticity $w$ and velocity field $u$, and then to physical space through IFFT. Simple operations such as multiplication and addition are performed according to conservation relations. As for the source term is characterized by a layer of S-Conv (*i.e.*, spectralConv2d_fast) with (width= 12, modes1= 12, modes2= 12).

- **Hybrid Operator.** For Lid2d and NSM2d datasets, the hybrid operator is chosen in spatial, while the RK-4 scheme is selected in temporal. According to the velocity part of the conservation equation, we set the kernel as five using trainable convolutional kernels as the convolution kernel parameters of diffusive and convective flows. We use a three-layer convolutional layer with channel 16 and GELU to represent the source term. Then, the intermediate results of the velocity field are fed into an S-Conv (width= 8, modes1= 8, modes2= 8) to map the complete velocity field and pressure field.

**ConvLSTM (Shi et al., 2015).** Specializing in spatial-temporal prediction, ConvLSTM blends LSTM's temporal cells with CNN spatial extraction. The setup consists of three distinct blocks: an encoding block employing a $5 \times 5$ convolution kernel with channel 32, an LSTM cell-based forecasting block, and a decoding block featuring 2 CNN layers with $5 \times 5$ kernels and 2 Res blocks. Notably, multi-step predictions are achieved through state concatenation within the forecasting block. Despite its strengths, its performance in complex process systems can be limited due to potential error accumulations.

**Dil-ResNet (Stachenfeld et al., 2021).** This model combines the encode-process-decode paradigm with the dilated convolutional network. The processor consists of $N = 4$ residual blocks connected in series, and each is made of dilated CNN stacks with residual connections. One stack consists of 7 dilated CNN layers with dilation rates of $(1, 2, 4, 8, 4, 2, 1)$, where a dilated rate of $N$ indicates that each pixel is convolved with multiples of $N$ pixels away. Each CNN layer in the processor is followed by ReLU activation. The key part of this network is a residual connection, which helps avoid vanishing gradients, and dilations allow long-range communication while preserving local structure. We found difficulties running this model on complex datasets due to computing and memory constraints.

**time-FNO2D (Li et al., 2020).** This model applies matrix multiplications in the spectral space with learnable complex weights for each component and linear updates and combines embedding in the spatial domain. The model consists of 2 MLP layers for encoding and decoding and 4 Fourier operation blocks (width= 12, modes1= 12, modes2= 12). Each block contains Fourier and CNN layers, followed by GELU activation. Optionally, low-pass filtering truncates high-frequency modes along each dimension in the Fourier-transformed grid.

**MIONet (Jin et al., 2022).** In the original paper, the Depth of MIONet is set to 2, the width is 200, and the number of parameters is 161K. Build MIONet with DeepXDE (Lu et al., 2021a) [3]. In this work, we have greatly adjusted the number of parameters; the number of parameters is about 20k, and the width is set to 20. Consistent with FNO, MLP is also introduced to construct the project layer for data, and then projection is also carried out in output. Other contents are consistent with the original text.

**U-FNet (Gupta & Brandstetter, 2022).** This model improves U-Net architectures, replacing lower blocks both in the downsampling and in the upsampling path of U-Net architectures by Fourier blocks, where each block consists of 2 FNO layers and residual connections. Other contents are consistent with the original text. In this work, we have adjusted $n\_input\_scalar\_components$ and $n\_output\_scalar\_components$ to the number of channels of the physical field in our datasets, and both $time\_history$ and $time\_future$ are set to 5 for better fitting.

**CNO (Raonić et al., 2023).** This model proposes a sequence of layers with the convolutional neural operator, mapping between bandlimited functions based on U-Net architectures. The convolutional neural operator consists of 4 different blocks, i.e., the downsampling block, the upsampling block, the invariant block, and the ResNet block. In the original paper, the width and height of spatial size for the mesh grid should be identical. In this work, we relaxed this restriction in activation function $filtered\_lrelu$ to fit on non-square grids like the NSM2d dataset. Other contents are consistent with the original text.

**PeRCNN (Rao et al., 2023).** The network consists of two components: a fully convolutional decoder as an initial state generator and a novel recurrent block named $\prod$-block for recursively updating the state variables. Here, since the available measurement size in our experiments is full, we have omitted this decoder. In the recurrent $\prod$-block, the state first goes through multiple parallel $1 \times 1$ Conv layers with stride 1 and output channel 32. The feature maps produced by these layers are then fused via the elementwise product operation. Then, the multi-channel goes through a conv layer with a filter size of 1 to obtain the output of the desired number of channels. We found this method unstable when approaching nonlinear complex terms and prone to NaN values during training.

**PPNN (Liu et al., 2022).** This model combines known partial nonlinear functions with a trainable neural network, which is named ConvResNet. The only difference between these two models is that a trainable portion of PPNN has an extra input variable $\mathcal{F}$, provided by the PDE-preserving portion

---

[3]https://github.com/lululxvi/deepxde

of PPNN. The state first goes through the decoder, which is made of four ConvResNet blocks, and each of them consists of a $7 \times 7$ kernel with 96 channels and a zero padding of 3. The following decoder includes a pixel shuffle with an upscale factor equal to 4 and a convolution layer with a $5 \times 5$ kernel. Due to the physics-aware design, this model shows lower relative error in the extrapolation range.

## A.7 ADDITIONAL EXPERIMENTAL RESULTS

In this section, we will first detail the data visualization and training and inference time cost-specific details, then show another ability of PAPM, adaptive resolution.

### A.7.1 VISUALIZATION

Fig. 7 and Fig. 8 showcase the results across five extrapolation time slices on Burgers2d and RD2d datasets. Both datasets clearly show that the physics-aware methods, PPNN and PAPM (our), can predict the dynamics of these two complex systems well. However, in the second half of the extrapolation ($T \geq \frac{1}{2}T_{end}$), It can be seen that our method PAPM is better than PPNN in local detail reconstruction. It is worth mentioning that our method has only 1% of the number of parameters and FLOPs of PPNN. However, the effect is better than PPNN, which further affirms the superiority of our structured design and specific spatio-temporal modeling method.

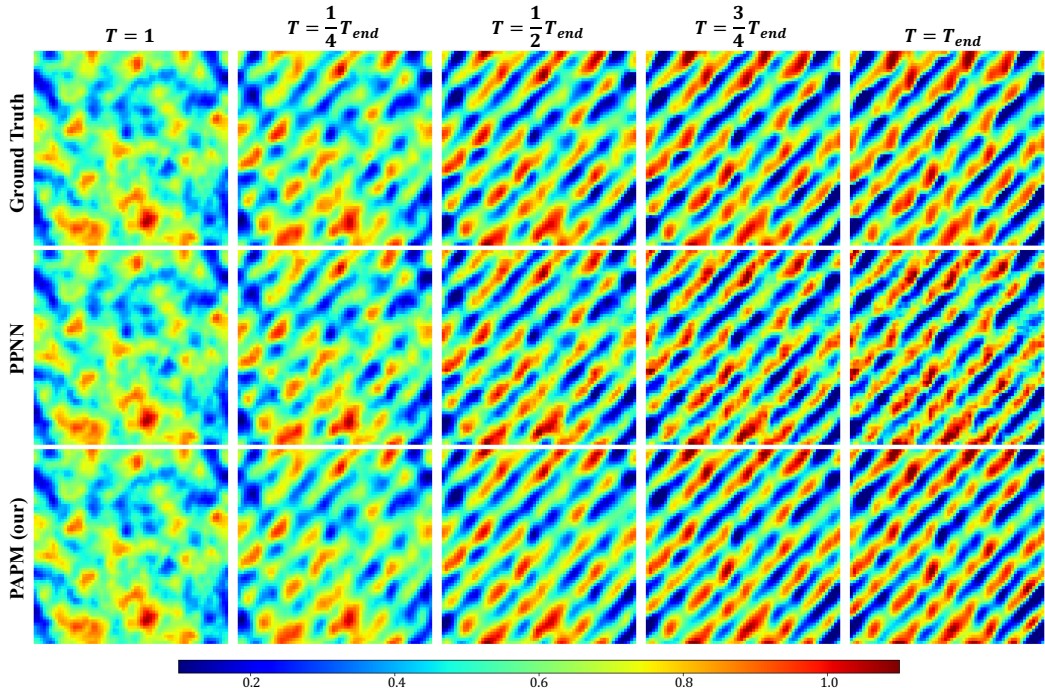

Figure 7: Predicted flow velocity ($\|\boldsymbol{u}\|_2$) snapshots by PPNN, and PAPM (Ours) vs. Ground Truth (GT) on Burgers2d dataset in T Ext. task.

### A.7.2 TRAINING AND INFERENCE TIME COST

Dataset generation for our work is notably resource-intensive, with inference costs ranging from $10^3 \sim 10^5$ s for public datasets and up to $10^6$ s for those we generated using COMSOL Multiphysics®. In stark contrast, both baselines and PAPM register inference times between $10^{-1} \sim 10$ s in Tab 5, achieving an improvement of 4 to 6 orders of magnitude. Notably, PAPM's time cost rivals or even surpasses baselines across different datasets. This indicates that introducing rigorous physical mechanisms, unlike traditional numerical algorithms, doesn't necessarily bloat time costs. PAPM's efficiency remains competitive with other data-driven methods.

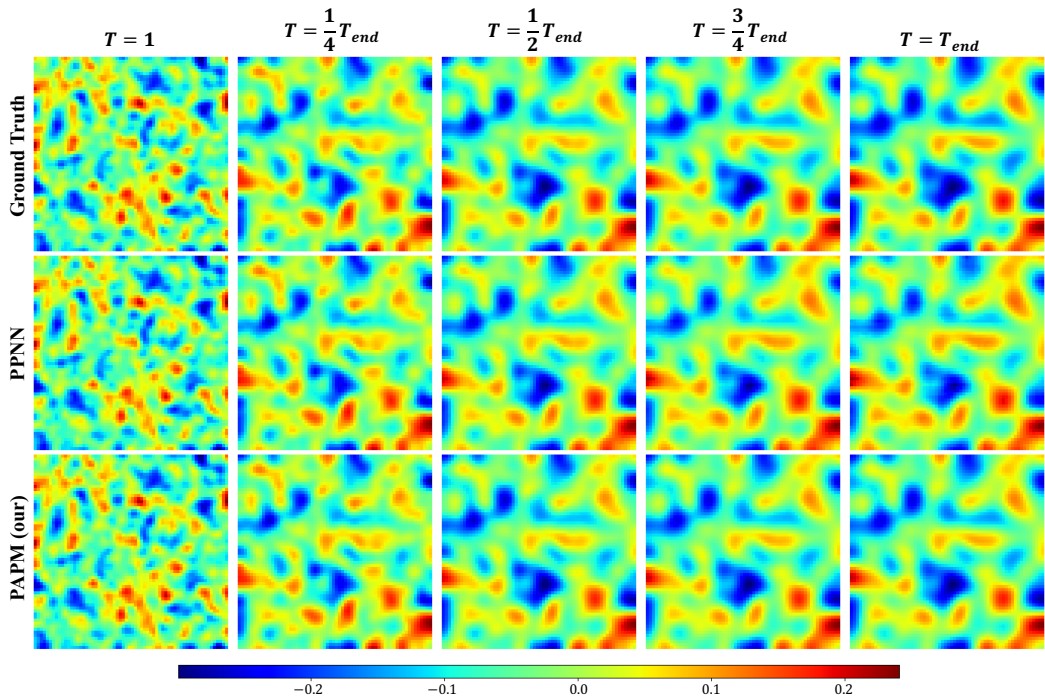

Figure 8: Predicted flow velocity ($\|u\|_2$) snapshots by PPNN, and PAPM (Ours) vs. Ground Truth (GT) on RD2d. dataset in C Ext. task.

Table 5: Training and inference time cost (iteration/second) of different baselines.

| Config | Burgers2d | | RD2d | | NS2d | | Lid2d | | NSM2d | |
|---|---|---|---|---|---|---|---|---|---|---|
| | Train | Infer | Train | Infer | Train | Infer | Train | Infer | Train | Infer |
| ConvLSTM | 5.41 | 1.12 | 21.41 | 3.94 | 7.05 | 0.86 | **8.68** | 3.22 | 4.39 | 0.92 |
| Dil-ResNet | 6.64 | 1.73 | 27.06 | 4.06 | 9.96 | 1.19 | 10.90 | 3.66 | 6.34 | 1.03 |
| time-FNO2D | 4.87 | 1.46 | 8.94 | 1.95 | 5.16 | 0.79 | 10.41 | 2.11 | **3.35** | 0.69 |
| MIONet | 5.69 | 1.58 | 8.69 | 2.03 | 5.05 | 0.89 | 10.54 | 3.02 | 4.03 | 0.76 |
| U-FNet | 3.64 | **0.52** | 14.56 | 1.96 | 6.67 | 0.51 | 10.42 | 1.14 | 6.96 | 0.82 |
| CNO | 4.02 | 0.60 | 15.72 | 2.28 | 4.92 | **0.44** | 11.08 | **1.12** | 5.90 | 0.68 |
| PeRCNN | 5.02 | 1.72 | **5.73** | 1.47 | 6.53 | 0.84 | 17.44 | 4.08 | 4.24 | 0.82 |
| PPNN | 5.07 | 0.96 | 8.88 | **1.19** | 4.87 | 0.91 | 15.58 | 3.44 | 8.08 | **0.64** |
| PAPM | **3.44** | 0.93 | 8.62 | 2.07 | **3.70** | 1.27 | 8.91 | 2.94 | 5.13 | 0.88 |

### A.7.3 Adaptive resolution

Incorporating prior physics knowledge into models brings a pivotal requirement: resolution independence akin to that observed in traditional numerical methods. This expectation guides our design of the Physics-Aware Proxy Model (PAPM). During its construction, we deliberate on two fundamental modes of matter motion, convection and diffusion, and the source term's impacts on model dynamics. Consequently, the different components of our model are designed to exhibit robustness against resolution variations. Specifically, in the convection and diffusion components (i.e., CF and DF in Fig. 2), the resolution scale (e.g., scale $\in [32, 64, 128, 256]$) serves as a hyperparameter influencing the convolution kernel parameters. Here, the mesh size is defined as $\frac{1}{scale}$, which, when combined with predefined kernels, imparts resolution awareness to the model. However, the original TSSM's ResNet-based source term lacks adaptability to resolution changes. To address this, we propose a modification in line with the lift-mapping-project structures seen in FNO (Li et al., 2020) and CNO (Raonić et al., 2023). We alter the ResNet structure into a lift-ResNet-project format for the localized operator's source term. In contrast, we adopt the S-Conv structure for spectral operators, aligning with the FNO approach.

As detailed in Table 6, we assess the adaptive resolution performance of various models, including our PAPM, across three public datasets: Burgers2d, RD2d, and NS2d (with $\nu = 1 \times 10^{-4}$). For each dataset, training was performed at a standard resolution of $[64, 64]$. We explored three scaling scenarios, $[1/2, 2, 4$, corresponding to resolutions of 32, 128, and 256, respectively, to conduct zero-shot evaluations of the models.

The results highlight that modifications to the PAPM structure have notably enhanced its adaptive resolution capabilities across different resolutions, confirming the effectiveness of this architecture. Notably, PAPM demonstrates robust performance in various scaling scenarios, indicating its resilience to resolution changes. Compared to other physical-aware methods, PAPM exhibits superior adaptive resolution ability. Moreover, the results of purely data-driven approaches, which lack this adaptive resolution capability, underscore the importance of integrating physics priors for enhanced adaptability to varying resolutions. However, it is also essential to acknowledge that regarding adaptive resolution, PAPM still lags behind other neural operators like FNO and CNO. This observation suggests a potential area for further improvement in PAPM's design and implementation.

Table 6: Main results ($\epsilon$) across different datasets in time extrapolation task with $0.5, 2, 4$ scaling.

| Config | Burgers2d | | | | | | | | RD2d | | | NS2d | | | |
| | C Int. | | | | C Ext. | | | | C Int. | | | $\nu = 1e$-4 | | | |
| Scale | 0.5 | 1 | 2 | 4 | 0.5 | 1 | 2 | 4 | 0.5 | 1 | 2 | 0.5 | 1 | 2 | 4 |
|---|---|---|---|---|---|---|---|---|---|---|---|---|---|---|---|
| ConvLSTM | 0.480 | 0.314 | 0.339 | 0.287 | 0.654 | 0.551 | 0.6581 | 0.679 | 0.879 | 0.815 | 0.900 | 0.894 | 0.877 | 0.923 | 0.925 |
| Dil-ResNet | 0.223 | 0.071 | 0.216 | 0.226 | 0.272 | 0.136 | 0.257 | 0.288 | 0.170 | 0.021 | 0.169 | 0.623 | 0.511 | 0.489 | 0.471 |
| time-FNO2D | 0.303 | 0.173 | 0.170 | 0.171 | 0.312 | 0.233 | 0.234 | 0.231 | 0.607 | 0.333 | 0.320 | **0.113** | 0.100 | **0.110** | **0.114** |
| U-FNet | 0.409 | 0.109 | 0.294 | 0.305 | 0.451 | 0.433 | 0.479 | 0.484 | 0.309 | 0.239 | 0.264 | 0.235 | 0.190 | 0.206 | 0.226 |
| CNO | 0.154 | 0.112 | 0.139 | 0.101 | 0.180 | 0.126 | 0.158 | 0.178 | 0.271 | 0.258 | 0.239 | 0.145 | 0.148 | 0.156 | 0.157 |
| PPNN | 0.263 | 0.047 | 0.242 | 0.246 | 0.329 | 0.132 | 0.306 | 0.295 | 0.442 | 0.030 | 0.491 | 0.891 | 0.357 | 0.543 | 0.591 |
| **Vanilla PAPM** | 0.144 | **0.039** | 0.084 | 0.107 | 0.205 | 0.101 | 0.140 | 0.164 | 0.064 | **0.018** | 0.079 | 0.141 | **0.097** | 0.117 | 0.133 |
| **PAPM** | **0.071** | **0.039** | **0.043** | **0.045** | **0.101** | **0.098** | **0.113** | **0.108** | **0.034** | **0.018** | **0.039** | 0.141 | **0.097** | 0.117 | 0.133 |

### A.8 Application of PAPM to Plasma Modeling.

Here, we consider a more complex process system, low-temperature argon plasma discharge[4], as depicted in Fig. 9. This process system is a classical multi-physics field coupling problem, which is a widely recognized phenomenon that occurs when applying an electric field to a gas, leading to gas ionization and plasma formation.

**Modeling.** The sequence of events involved in this process includes the creation of free electrons, ionization, electron impact excitation, and recombination, which can be described mathematically using mechanism equations, such as the maxwell, drift-diffusion, and fluid equations. It does not strictly adhere to the definition in Eq. 1. Still, for the physical quantities to be modeled, their

---

[4]More detailed information can be shown in https://www.comsol.com/model/gec-icp-reactor-argon-chemistry-8649.

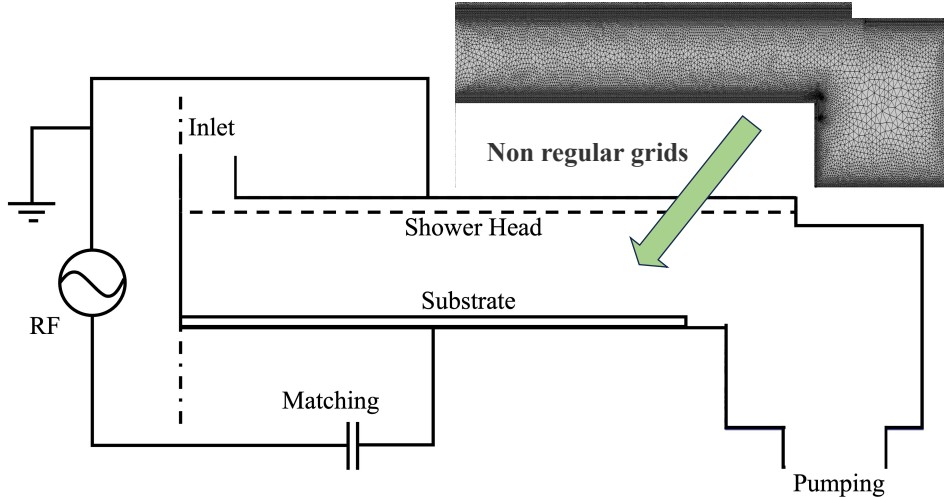

Figure 9: The schematic of 2D low-temperature Argon plasma discharge with non-regular grids.

overall pattern follows the basic setting in which convection, diffusion, and source terms act together. Therefore, we can still construct a proxy model for plasma through PAPM.

**Data generation.** We implemented this complex example by COMSOL®. Here, the input condition is not the initial conditions and equation coefficients but the working condition parameters. The parameter range of $\eta$ is defined as $[V_{min}, V_{max}] \times [P_{min}, P_{max}]$, which contains two parameters for voltage and air pressure. The dynamic of the system is completely different, with different parameters. For instance, as illustrated in Fig. 10, the significant variations observed in the dynamic evolutionary characteristics among electron density, electron temperatures, and potentials under different simulation conditions emphasize the distinctiveness of their physical attributes.

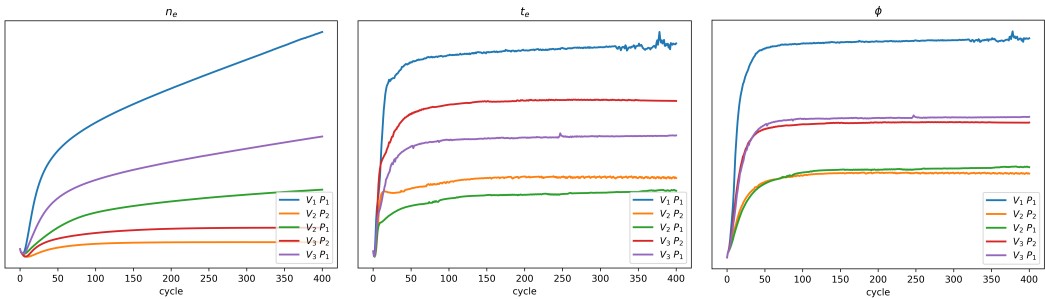

Figure 10: Variation characteristics of electron density $n_e$, electron temperature $t_e$, and electric potential $\phi$. Dynamic evolution curves of the mean value from the initial to the final state under different simulation conditions.

Our task is to model the plasma dynamics within a set range of operating conditions, the same as the previous task setting. In particular, for the working condition of each group $\eta_i = [V_i, P_i]$, input model $S_{t_0} = [s_0, \ldots, s_{t_0-1}]$, a total of $t_0$ step state, we want to get the state of the following $T_{end} - t_0$ step. Here, $s \in \mathcal{R}^3$ is a vector, where $s = [n_e, t_e, \phi]$, denoting electron density, electron temperature, and electric potential, respectively. In training, we only the first half data to training, input $S_{t_0}$, label data for $[s_{t_0} - 1, \ldots, s_{T_{end}/2}]$. We perform **five-fold cross-validation** for plasma data with 40 sets of uniformly sampled data under the operating range setting of $[150, 500] \times [20, 70]$.

**Results.** For handling irregular mesh data, we initially applied bilinear interpolation to transform it into regular mesh data with a spatial resolution of $[160, 160]$. We set the initial time $t_0 = 5$ and

the end time $T_{end} = 200$, using data from the first half of the $T_{end}/2$ time steps for training. In the testing phase, we input the initial state sequence at $t_0 = 5$ to predict the states from $t_0$ to $T_{end}$.

Despite the absence of exact physics equations for the plasma process, we effectively derived gradient information of three physical quantities through convection and diffusion components. This was followed by feature fusion via the source component, aiding in understanding the potential underlying mechanism equation. The RK-4 format was employed for time stepping. As demonstrated in Table 7, our method surpasses others in accuracy, generalization, and efficiency. Moreover, we explored enhancing the model's complexity by scaling up the number of parameters in PAPM. This involved varying the hidden channel numbers (hidden_channel = $[16, 32, 64]$), which led to a significant increase in the parameter count (from 3w to 20w). However, the improvements in model performance were relatively modest when weighed against the substantial increase in computational resources incurred. This discrepancy might be attributed to the limitations posed by the size of the dataset.

Table 7: Main results ($\epsilon$), FLOPs, training and inference time cost (iteration/second), and the number of trainable parameters ($N_P$) on Plasma2d.

| Config | $\epsilon$ | FLOPs | Train | Test | $N_p$ |
|--------|-----------|-------|-------|------|-------|
| convLSTM | 0.750 | 2.05G | 5.05 | 0.43 | 0.080M |
| Dil-ResNet | 0.316 | 3.90G | 7.61 | 0.59 | 0.152M |
| timeFNO2d | 0.286 | 0.11G | **2.37** | **0.31** | 0.465M |
| U-FNet | 0.718 | 5.22G | 9.38 | 1.49 | 10.091M |
| CNO | 0.407 | 3.50G | 8.46 | 0.92 | 2.674M |
| PPNN | 0.228 | 3.01G | 4.60 | 0.63 | 1.300M |
| **PAPM-16** | 0.178 | **0.07G** | 4.66 | 0.47 | **0.032M** |
| **PAPM-32** | 0.176 | 2.05G | 6.94 | 0.60 | 0.082M |
| **PAPM-64** | **0.171** | 6.41G | 9.94 | 0.79 | 0.245M |

## A.9  LIMITATION AND FUTURE WORK

Our approach centers around two primary considerations. Firstly, we aim to extend our model to more realistic process systems, particularly those used in industrial simulations. While our current validation has been on standard 2D spatio-temporal dynamic systems with well-defined process model mechanisms, PAPM has demonstrated a superior balance in accuracy, operational efficiency, and generalization capabilities. A notable example is our application of PAPM to plasma, a complex case involving multi-physics field coupling. The results from this plasma experiment exemplify PAPM's strengths in handling more intricate process systems. Moving forward, we plan to explore PAPM's architecture in multi-physics field coupling scenarios, such as fluid-structure and thermal fluid-structure coupling problems.

The second consideration is the potential scalability of PAPM concepts to a larger model framework. Since most dynamic systems adhere to three primary aspects – convection (with advection being a specific case), diffusion, and source terms – we are keen to integrate this structured design into developing larger-scale models. We aim to construct a foundational model for process systems that can efficiently and accurately model various processes with a unified approach.

