# OpenReview forum: "PAPM: A Physics-aware Proxy Model for Process Systems"
_ICLR.cc/2024/Conference — ICLR 2024 Conference Withdrawn Submission_

### Official Review · Reviewer_9hXa · 2023-10-31

**Soundness:** 2 fair
**Presentation:** 2 fair
**Contribution:** 2 fair
**Rating:** 3
**Confidence:** 4

**Summary:**

This paper introduces PAPM, a spatio-temporal model to capture complex dynamics which arguably follow similar patters, i.e., a mixture of diffusion and/or convection flows, internal and external source terms. The PAPM architecture encodes the state of the system, and depending on the problem at hand applies either localized, spectral, or hybrid operators to parameterize the different operators. Subsequently, time-stepping schemes are applied to mimic temporal updates. PAPM is tested on 4 known 2D fluid mechanics benchmarks systems.

**Strengths:**

- Introducing parameterized operators is a very interesting contribution.

**Weaknesses:**

- The presentation is slightly hard to follow, it is not clear to me how exactly all these operators are parameterized and how such models can be scaled up. Is there only one operator block used or can these modules be stacked? Pseudocode / real code would definitely help.
- The models are evaluated on a fixed grid with fixed resolution. For such systems standard models such as modern U-Nets and / or convolutional based neural operators should be used for comparison (Raonic et al, Gupta et al), or even Vision Transformers. An alternative is to showcase resolution independency to justify the comparisons.
- I am pretty puzzled by the low number of parameters. It seems that hardly any model uses more than 1 million parameters. This is in my opinion a heavy under-parameterization for 2D problems. Compare for example Fig 1 in Gupta et al?
- The paper makes a strong claim for better physics modeling, i.e., strong physics bias, yet there is no evidence that with low number of samples the performance is better compared to baseline models.
- Figure 6 is not comparing to the best baseline model but FNO which has 10 times worse performance than Dilated ResNets on the RD2d task.
- It is impossible to judge how the individual components contribute to the results - ablation would help.


Raonić, B., Molinaro, R., Rohner, T., Mishra, S., & de Bezenac, E. (2023). Convolutional Neural Operators. arXiv preprint arXiv:2302.01178.

Gupta, Jayesh K., and Johannes Brandstetter. "Towards multi-spatiotemporal-scale generalized pde modeling." arXiv preprint arXiv:2209.15616 (2022).

**Questions:**

- How can PAPM be extended to variable grid sizes, or to non regular grids?
- How can PAPM be scaled up to larger number of parameters?
- Would it be possible to resort to the standard terminology of "operator learning" which is now standard in the community?

---

> ### Author Response · Authors · 2023-11-19
> **Response to Reviewer 9hXa (1)**
>
> Thank you for your valuable feedback, which has been instrumental in refining our paper. We have carefully considered your suggestions and have made corresponding adjustments to enhance the clarity and depth of our work.
>
> ## 1. Clearer Model Presentation
> - 1.1 **Motivation**
>     - We appreciate your interest in our model's motivation. In the field of physics-informed machine learning (PIML), our approach follows the trend of integrating physical knowledge into the model structure. This approach, seen in recent works like PDE-Net [1], FINN [2], PPNN [3], PiNDiff [4] and PeRCNN [5], allows us to embed prior physics dynamics, such as PDEs, to guide our model's structural design and enhance inductive biases. This integration not only boosts the model’s ability to understand dynamic mechanisms for better nonlinear approximation but also significantly trims the number of required parameters. Consequently, we achieve enhanced accuracy and generalization, even with limited data. For instance, PeRCNN, with less than 1k parameters, effectively models diffusion equations.
>     - In contrast, data-driven operator learning models, such as FNO [6], MIONet [7], U-FNet [8] and CNO [9], that rely on increasing the number of parameters can improve nonlinear approximation. But it tends to overfit with limited data, reducing generalization. For example, as demonstrated in **Fig. 5** of our paper, FNO performs well on test sets matching the training data's step size, but its efficacy declines sharply with time extrapolation. Therefore, incorporating more prior physics into the model's structural design emerges as a superior strategy. However, it's important to note that the mechanism information we integrate is incomplete, which precludes the use of loss function embedding in this context.
>
> - 1.2 **Model Design**
>     - Our paper follows this line (**integrating physical knowledge into the model structure**), focusing on operator learning tasks under process model. We have detailed the structured network design (see **Fig.2**) and spatio-temporal stepping method (see **Fig.3**) to provide a comprehensive view. The inclusion of pseudocode (see **Appendix A.4**) and the **source code** (see **supplementary material**) is intended to facilitate better understanding.
>
> ## 2. More Comprehensive Experiments
>
> - 2.1 Comparison with Existing Methods
>     - In the revised version, we introduced two new baseline models: U-FNets [1] and Convolutional Neural Operators (CNO) [2], as demonstrated in **Table 1**(below). These additions enrich our comparative analysis, allowing us to explore PAPM's performance in relation to baseline models in terms of accuracy, parameter count, computational speed, and data efficiency in **Table 2**(below). PAPM exhibits the most balanced trade-off between parameter count and performance among all methods evaluated, from explicit structures (Burgers2d, RD2d) to implicit (NS2d) and more complex hybrid structures (Lid2d, NSM2d). Notably, even though PAPM utilizes only 1\% of the parameters employed by the prior leading method, PPNN, it still outperforms it by a large margin. In a nutshell, our model enhances the performance by an average of 6.4\% over nine tasks, which affirms PAPM as a versatile and efficient framework suitable for diverse process systems. For the details of the above five datasets, please refer to **Tab. 2,3** and **Fig. 4,5,6** in the revised paper.
> - 2.2 Adaptive resolution experiments
>     -  We have also included adaptive resolution experiments across different resolutions on three datasets, highlighting PAPM's adaptive resolution capabilities. Incorporating physics-based knowledge into our models is crucial for achieving resolution independence. In **Appendix A.7.3**, we delve into the adaptive resolution capabilities of PAPM and our baseline models. We also discuss potential enhancements (see **Table 3 PAPM**, below) based on the concept of operator learning, like expanding the channel domain through lift and project functions to improve model adaptability. As demonstrated in Tab.1, The burgers2d dataset is used as an example to show the results of different baselines and PAPM under four resolutions (scale = $0.5$, $1$, $2$, $4$, The corresponding resolution is $[32, 64, 128,  256]$). The results highlight that modifications to the PAPM structure have notably enhanced its adaptive resolution capabilities across different resolutions, confirming the effectiveness of this architecture. PAPM demonstrates robust performance in various scaling scenarios,  indicating its resilience to resolution changes. Compared to other physical-aware methods, PAPM exhibits superior adaptive resolution ability. Moreover, the results of purely data-driven approaches,  which lack this adaptive resolution capability, underscore the importance of integrating physics priors for enhanced adaptability to varying resolutions. For more details, please refer to **Appendix A.7.3**.

---

> > ### Author Response · Authors · 2023-11-19
> > **Response to Reviewer 9hXa (2)**
> >
> > **Table 1**: Main results $\epsilon$ across different datasets in time extrapolation task.
> >
> > | **Config**      | **Burgers2d C Int.** | **Burgers2d C Ext.** | **RD2d C Int.** | **NS2d $\nu$=1e-3** | **NS2d $\nu$=1e-4** | **NS2d $\nu$=1e-5** | **Lid2d C Ext.** | **NSM2d C Int.** | **NSM2d C Ext.** |
> > |-----------------|---------------------:|---------------------:|----------------:|--------------------:|--------------------:|--------------------:|-----------------:|-----------------:|-----------------:|
> > | ConvLSTM [10]    |                0.314 |                0.551 |           0.815 |               0.781 |               0.877 |               0.788 |            1.323 |             0.910 |            1.102 |
> > | Dil-ResNet [11]  |                0.071 |                0.136 |           0.021 |               0.152 |               0.511 |               0.199 |            0.261 |             0.288 |            0.314 |
> > | time-FNO2D [6]  |                0.173 |                0.233 |           0.333 |               0.118 |              0.100  |               0.033 |            0.265 |             0.341 |            0.443 |
> > | MIONet [7]      |                0.181 |                0.212 |           0.247 |               0.139 |               0.114 |               0.051 |            0.221 |             0.268 |            0.440 |
> > | *U-FNet* [8]    |             *0.109*  |             *0.433*  |           *0.239* |           *0.191* |             *0.190* |             *0.256* |           *0.192* |           *0.257* |         *0.457* |
> > | *CNO* [9]       |             *0.112*  |             *0.126*  |           *0.258* |           *0.125* |             *0.148* |           **0.030** |           *0.218* |            *0.197* |        *0.355* |
> > | PeRCNN [5]      |                0.212 |                0.282 |           0.773 |               0.571 |               0.591 |               0.275 |            0.534  |             0.493 |           0.493 |
> > | PPNN [3]        |                0.047 |                0.132 |           0.030 |               0.365 |               0.357 |               0.046 |            0.163  |             0.206 |           0.264 |
> > | **PAPM (Our)**  |            **0.039** |            **0.101** |         **0.018** |           **0.110** |         **0.097** |               0.034 |          **0.160** |           **0.189** |    **0.245** |
> >
> >
> > **Table 2**: Main results ($\epsilon$), FLOPs, and comparison of the number of trainable parameters ($N_P$) on Lid2d.
> > | **Config** | **FLOPs** | **$N_p$** | **$\epsilon$** |
> > | ---------- | --------- | ----------- | --------------- |
> > | convLSTM   | 327.54M   | 211k        | 1.323           |
> > | Dil-ResNet | 624.00M   | 152k        | 0.261           |
> > | timeFNO2d  | 6.89M     | 464k        | 0.265           |
> > | MIONet     | 6.89M     | 261k        | 0.221           |
> > | U-FNet     | 559.89M   | 9854k       | 0.192           |
> > | CNO        | 835.37M   | 2612k       | 0.218           |
> > | PeRCNN     | 3.44M     | 1k          | 0.534           |
> > | PPNN       | 348.56M   | 1213k       | 0.163           |
> > | **PAPM**   | **1.22M** | **11k**     | **0.160**       |
> >
> >
> > **Table 3**: Main results $\epsilon$ in Burgers2d dataset for time extrapolation task with $[0.5, 1, 2, 4]$ scaling.
> > | **Config**         | **Burgers2d 0.5** | **Burgers2d 1** | **Burgers2d 2** | **Burgers2d 4** |
> > | ------------------ | ----------------: | --------------: | --------------: | --------------: |
> > | ConvLSTM           |             0.480 |           0.314 |           0.339 |           0.287 |
> > | Dil-ResNet         |             0.223 |           0.071 |           0.216 |           0.226 |
> > | time-FNO2D         |             0.303 |           0.173 |           0.170 |           0.171 |
> > | U-FNet             |             0.409 |           0.109 |           0.294 |           0.305 |
> > | CNO                |             0.154 |           0.112 |           0.139 |           0.101 |
> > | PPNN               |             0.263 |           0.047 |           0.242 |           0.246 |
> > | **Vanilla PAPM**   |             0.144 |     **0.039**   |           0.084 |           0.107 |
> > | **PAPM**           |       **0.071**   |     **0.039**   |     **0.043**   |       **0.045** |

---

> ### Author Response · Authors · 2023-11-19
> **Response to Reviewer 9hXa (3)**
>
> ## 3. Analysis on Data Efficiency
> - To evaluate data efficiency, we conducted tests using RD2d dataset (see [12], this dataset can be downloaded at https://github.com/pdebench/PDEBench) as a representative example, with Dil-Resnet and PPNN symbolizing pure data-driven and physics-aware methods (see **Table 4** below). The results, displayed in **Fig.6**, depict PAPM's efficiency concerning data volume and label data step size in training.
>     - 3.1 **Amount of Data**: With a fixed 20% reserved for the test set, the remaining 80% of the total data is allocated to the training set. We systematically varied the training data volume, ranging from initially utilizing only 5% of the training set and progressively increasing it to the entire 100%. PAPM's relative error distinctly outperforms other baselines, **especially with limited data (5%)**. As depicted in **Fig.5(Left)**, PAPM's error consistently surpasses other methods, stabilizing below 2% as the training data volume increases.
>     - 3.2 **Time Step Size**: We varied the data step size from $1/10$ to half of the total, increasing in tenths. Results of  **Fig.5(Right)** reveal that PAPM can achieve long-range time extrapolation with minimal dynamic steps, consistently outshining other methods even with shorter training data step sizes.
>
> **Table 4: Main results $\epsilon$ in RD2d dataset with time extrapolation task.**
>
> | **Config**      | **RD2d** |
> |-----------------|----------------:|
> | ConvLSTM [10]   |           0.815 |
> | Dil-ResNet [11] |           0.021 |
> | time-FNO2D [6]  |           0.333 |
> | MIONet [7]      |           0.247 |
> | *U-FNet* [8]    |         *0.239* |
> | *CNO* [9]       |        *0.112*  |
> | PeRCNN [5]      |           0.212 |
> | PPNN [3]        |           0.030 |
> | **PAPM (Our)**  |       **0.018** |
>
> ## 4. Ablation Study
> - **Setting.** We selected the Burger2d dataset due to its representation of diffusion, convection, and force terms. Several configurations are defined to determine the effects of individual components. **no_DF** excludes diffusion, while **no_CF** omits convection. **no_Phy** retains only a structure with a residual connection, eliminating both diffusion and convection. The **no_BCs** setup removes explicit BCs embedding, **no_All** is purely data-driven, and **no_Iter** bypasses the Iterative Refinement Rounds training strategy.
> - **Results.**  As demonstrated in **Table 5** (below), key findings include recognizing the crucial roles of diffusion and convection in dictating system dynamics. This was evident when the no_DF configuration showed that integrating the viscosity coefficient with the diffusion term was vital. Its absence led to significant errors, most notably in parameter extrapolation tasks. The necessity of boundary adherence to physical laws became clear with the no_BCs approach, as it notably reduced BC's relative errors. Lastly, the no_Iter setup accentuated the importance of the causal training strategy in the model's training process.
>
> **Table 5: Comparison of the main result ($\epsilon$) and the number of trainable parameters ($N_P$) on Burgers2d.**
>
> | **Config**  | **\$N_P\$** | **C Int. \$\epsilon\$** | **C Int. BC \$\epsilon\$** | **C Ext. \$\epsilon\$** | **C Ext. BC \$\epsilon\$** |
> | ----------- | ----------- | ----------------------- | -------------------------- | ----------------------- | -------------------------- |
> | no_DF       | 13.90k      | 0.067                   | 0.051                      | 0.207                   | 0.067                      |
> | no_CF       | 13.93k      | 0.062                   | 0.043                      | 0.131                   | 0.054                      |
> | no_Phy      | 13.84k      | 0.149                   | 0.051                      | 0.210                   | 0.144                      |
> | no_BCs      | 13.99k      | 0.068                   | 0.097                      | 0.136                   | 0.193                      |
> | no_All      | 13.84k      | 0.162                   | 0.195                      | 0.216                   | 0.250                      |
> | no_Iter     | 13.99k      | 0.080                   | 0.039                      | 0.141                   | 0.048                      |
> | **PAPM**    | **13.99k**  | **0.039**               | **0.037**                  | **0.101**               | **0.043**                  |

---

> ### Author Response · Authors · 2023-11-19
> **Response to Reviewer 9hXa (4)**
>
> ## 5. More complex Applications:
> - **Q**: How can PAPM be extended to variable grid sizes, or to non regular grids?
> - **A**: PAPM be extended to variable grid sizes can see in **Adaptive resolution experiments** (see **2.2** above). And to non regular grids as follows:
>     - **a**. Using bilinear interpolation to convert non-uniform grids to standard grids, minimizing errors especially in dense grid scenarios.
>     - **b**. Employing a lift mapping function, similar to the geo-FNO [13] strategy, to transform non-uniform meshes into uniform ones.
>
>     In our case study of a low-temperature argon plasma discharge (see **Fig. 9**), we chose the first approach due to the high density of the simulation grid, allowing for effective direct interpolation before training the PAPM model. We explore a more complex example of low-temperature argon plasma discharge, which presents a challenging scenario with non-regular grids and complex mechanisms. The mechanism equation of this process is far more complex than Eq. 1 and 2, involving the coupling of multiple physical fields such as the electric field, temperature field, and pressure field. It does not strictly adhere to the definition in Eq. 1. Still, for the physical quantities to be modeled, their overall pattern follows the primary setting in which convection, diffusion, and source terms act together. Therefore, we can still construct a proxy model for plasma through PAPM. As demonstrated in **Table 6**(below), our method surpasses others in accuracy, generalization, and efficiency. For A more detailed introduction, please refer to **Appendix A.8**. We will use this example to show the generalization ability of PAPM further.
>
> **Table 6: Main results ($\epsilon$), FLOPs, training and inference time cost (iteration/second), and the number of trainable parameters ($N_P$) on Plasma2d.**
> | **Config**    | **\$\epsilon\$** | **FLOPs**  | **Train** | **Test** | **$N_p$**   |
> | ------------- | ---------------- | ---------- | --------- | -------- | ------------- |
> | convLSTM [10]      | 0.750            | 2.05G      | 5.05      | 0.43     | 0.080M        |
> | Dil-ResNet [11]    | 0.316            | 3.90G      | 7.61      | 0.59     | 0.152M        |
> | timeFNO2d [6]     | 0.286            | 0.11G      | **2.37**      | **0.31**     | 0.465M        |
> | U-FNet [8]        | 0.718            | 5.22G      | 9.38      | 1.49     | 10.091M       |
> | CNO [9]           | 0.407            | 3.50G      | 8.46      | 0.92     | 2.674M        |
> | PPNN [3]          | 0.228            | 3.01G      | 4.60      | 0.63     | 1.300M        |
> | **PAPM**      | **0.178**            | **0.07G**      | 4.66      | 0.47     | **0.032M**   |
>
> ## 6. Additional Experiments on Scaling Up Parameters
> - **Q**: How can PAPM be scaled up to larger number of parameters?
> - **A**: In terms of scaling up the number of parameters, we have experimented with different configurations, particularly in the source term. For example, in the plasma experiment, we expanded the parameter quantity of PAPM, considering varying hidden channel numbers (hidden\_channel = $[16, 32, 64]$,), which significantly increased the parameter count (parameters from 3w to 20w), as demonstrated in **Table 7** (below). However, the improvements in model performance were relatively modest when weighed against the substantial increase in computational resources incurred. This discrepancy might be attributed to the limitations posed by the size of the dataset.
>
> **Table 7: Main results ($\epsilon$), FLOPs, training and inference time cost (iteration/second), and the number of trainable parameters (\$N_P\$) on Plasma2d.**
>
> | **Config**    | **$\epsilon$** | **FLOPs**  | **Train** | **Test** | **$N_p$**   |
> | ------------- | ---------------- | ---------- | --------- | -------- | ------------- |
> | convLSTM [10]    | 0.750            | 2.05G      | 5.05      | 0.43     | 0.080M        |
> | Dil-ResNet [11]   | 0.316            | 3.90G      | 7.61      | 0.59     | 0.152M        |
> | timeFNO2d [6]    | 0.286            | 0.11G      | **2.37**      | **0.31**     | 0.465M        |
> | U-FNet [8]       | 0.718            | 5.22G      | 9.38      | 1.49     | 10.091M       |
> | CNO [9]          | 0.407            | 3.50G      | 8.46      | 0.92     | 2.674M        |
> | PPNN [3]         | 0.228            | 3.01G      | 4.60      | 0.63     | 1.300M        |
> | **PAPM-16**   | 0.178            | **0.07G**      | 4.66      | 0.47     | **0.032M**        |
> | **PAPM-32**   | 0.176            | 2.05G      | 6.94      | 0.60     | 0.082M        |
> | **PAPM-64**   | **0.171**            | 6.41G      | 9.94      | 0.79     | 0.245M        |

---

> ### Author Response · Authors · 2023-11-19
> **Response to Reviewer 9hXa (5)**
>
> ## 7. Terminology Choice
> - **Q**: Would it be possible to resort to the standard terminology of "operator learning" which is now standard in the community?
> - **A**: Regarding the terminology, we have opted for 'physical-aware proxy model' and 'spatiotemporal stepping method' to align more closely with our research approach. This choice reflects our focus on embedding physical knowledge into the model structure.
>
> We hope these responses adequately address your concerns and demonstrate our commitment to advancing research in the PIML field. We are grateful for your constructive feedback and remain open to further discussions.
>
> ## **Reference**
> - [1] Long Z, Lu Y, Ma X, et al. Pde-net: Learning pdes from data[C]//International conference on machine learning. PMLR, 2018: 3208-3216.
> - [2] Karlbauer M, Praditia T, Otte S, et al. Composing partial differential equations with physics-aware neural networks[C]//International Conference on Machine Learning. PMLR, 2022: 10773-10801.
> - [3] Liu X Y, Sun H, Zhu M, et al. Predicting parametric spatiotemporal dynamics by multi-resolution PDE structure-preserved deep learning[J]. arXiv preprint arXiv:2205.03990, 2022.
> - [4] Akhare D, Luo T, Wang J X. Physics-integrated neural differentiable (PiNDiff) model for composites manufacturing[J]. Computer Methods in Applied Mechanics and Engineering, 2023, 406: 115902.
> - [5] Rao C, Ren P, Wang Q, et al. Encoding physics to learn reaction–diffusion processes[J]. Nature Machine Intelligence, 2023, 5(7): 765-779.
> - [6] Li Z, Kovachki N B, Azizzadenesheli K, et al. Fourier Neural Operator for Parametric Partial Differential Equations[C]//International Conference on Learning Representations. 2020.
> - [7] Jin P, Meng S, Lu L. MIONet: Learning multiple-input operators via tensor product[J]. SIAM Journal on Scientific Computing, 2022, 44(6): A3490-A3514.
> - [8] Gupta J K, Brandstetter J. Towards multi-spatiotemporal-scale generalized pde modeling[J]. arXiv preprint arXiv:2209.15616, 2022.
> - [9] Raonić B, Molinaro R, Rohner T, et al. Convolutional Neural Operators[J]. arXiv preprint arXiv:2302.01178, 2023.
> - [10] Shi X, Chen Z, Wang H, et al. Convolutional LSTM network: A machine learning approach for precipitation nowcasting[J]. Advances in neural information processing systems, 2015, 28.
> - [11] Stachenfeld K, Fielding D B, Kochkov D, et al. Learned coarse models for efficient turbulence simulation[J]. arXiv preprint arXiv:2112.15275, 2021.
> - [12] Takamoto M, Praditia T, Leiteritz R, et al. PDEBench: An extensive benchmark for scientific machine learning[J]. Advances in Neural Information Processing Systems, 2022, 35: 1596-1611.
> - [13] Li Z, Huang D Z, Liu B, et al. Fourier neural operator with learned deformations for pdes on general geometries[J]. arXiv preprint arXiv:2207.05209, 2022.

---

> > ### Comment · Reviewer_9hXa · 2023-11-21
> > **Post-rebuttal**
> >
> > Dear authors,
> > thanks to the authors for the rebuttal.
> >
> > However, I got the impression that most of the answers are quite generic and repetitive amongst reviewers. My most important concern, i.e., that baseline models were used with unusual low number of parameters, was unfortunately ignored in the rebuttal. Since I did not get any response on this issue, I am keeping my score.

---

> > > ### Author Response · Authors · 2023-11-21
> > > **Response to "low number of parameters" concern (1).**
> > >
> > > ## 1. Data-driven models need more parameters to fit the data.
> > >
> > > For data-driven models, it is expected to need a model with a large enough number of parameters to fit the number. As you said, in most cases, a data-driven model usually uses more than 1 million parameters for 2D problems. For example, as shown in **Table 1**, we found this phenomenon in our previous experiments, we adjusted the parameter amount of **U-FNet** model on **RD2d** dataset, and changed the **hidden_channels** from 2 to 16. Its parameter amount also increased from 0.159M to 9.853M, gradually improving the effect. However, increasing the model parameter amount further (hidden_channels=20) decreases the outcome.
> > >
> > > In addition, as shown in **Table 2**, when we model Lid2D data, the small parameter models **convLSTM** and **Dil-ResNet** in the data-driven method perform worse than the large parameter models such as **U-FNet** and **CNO**. These conclusions are verified in our experiments.
> > >
> > > **Table 1: Main results ($\epsilon$) and the number of trainable parameters ($N_P$) on RD2d dataset.**
> > > | **Config**  | **$\epsilon$** | **$N_p$**  |
> > > | ------------| ---------------| -----------|
> > > | U-FNet-20   | 0.245          | 15.764M    |
> > > | U-FNet-16   | 0.239          | 9.853M     |
> > > | U-FNet-12   | 0.258          | 5.678M     |
> > > | U-FNet-8    | 0.314          | 2.525M     |
> > > | U-FNet-4    | 0.427          | 0.633M     |
> > > | U-FNet-2    | 0.652          | 0.159M     |
> > >
> > > **Table 2:  Main results ($\epsilon$) and the number of trainable parameters ($N_P$) of data-driven methods on Lid2d dataset.**
> > > | **Config** | **$N_p$** | **$\epsilon$** |
> > > | ---------- | ----------- | -------------|
> > > | convLSTM   | 211k        | 1.323        |
> > > | Dil-ResNet | 152k        | 0.261        |
> > > | timeFNO2d  | 464k        | 0.265        |
> > > | MIONet     | 261k        | 0.221        |
> > > | U-FNet     | 9854k       | 0.192        |
> > > | CNO        | 2612k       | 0.218        |
> > >
> > > ## **2. The more physics, the less parameters.**
> > >
> > > However, when we consider adding more prior physics to the model, the model no longer requires many parameters. In [1, 2, 3, 4] these review articles, a common point is emphasized: in the case of a fixed amount of data, **the more physical information is introduced into the neural network model, the fewer parameters are required for the model**. On the contrary, if there is no prior physics, the model needs more parameters to fit the data.
> > >
> > > This view has also been fully verified in recent works like PDE-Net [5], FINN [6], PPNN [7], PiNDiff [8], and PeRCNN [9] in 2D problems. Embedding more physical prior knowledge into the model, such as boundary conditions and PDE, to guide our model's structural design. This can significantly increase the **inductive biases** of the model, thereby **reducing** the number of parameters utilized in these models and our models, PAPM. For instance, PeRCNN, with less than 1k parameters, effectively models three complex 2D diffusion equations. Our approach follows the trend of **integrating prior physics into the model structure**.
> > >
> > > ## **Reference**
> > > - [1] Karniadakis G E, Kevrekidis I G, Lu L, et al. Physics-informed machine learning[J]. Nature Reviews Physics, 2021, 3(6): 422-440.
> > > - [2] Cuomo S, Di Cola V S, Giampaolo F, et al. Scientific machine learning through physics–informed neural networks: Where we are and what’s next[J]. Journal of Scientific Computing, 2022, 92(3): 88.
> > > - [3] Tu H, Moura S, Wang Y, et al. Integrating physics-based modeling with machine learning for lithium-ion batteries[J]. Applied Energy, 2023, 329: 120289.
> > > - [4] Lu L, Meng X, Mao Z, et al. DeepXDE: A deep learning library for solving differential equations[J]. SIAM review, 2021, 63(1): 208-228.
> > > - [5] Long Z, Lu Y, Ma X, et al. Pde-net: Learning pdes from data[C]//International conference on machine learning. PMLR, 2018: 3208-3216.
> > > - [6] Karlbauer M, Praditia T, Otte S, et al. Composing partial differential equations with physics-aware neural networks[C]//International Conference on Machine Learning. PMLR, 2022: 10773-10801.
> > > - [7] Liu X Y, Sun H, Zhu M, et al. Predicting parametric spatiotemporal dynamics by multi-resolution PDE structure-preserved deep learning[J]. arXiv preprint arXiv:2205.03990, 2022.
> > > - [8] Akhare D, Luo T, Wang J X. Physics-integrated neural differentiable (PiNDiff) model for composites manufacturing[J]. Computer Methods in Applied Mechanics and Engineering, 2023, 406: 115902.
> > > - [9] Rao C, Ren P, Wang Q, et al. Encoding physics to learn reaction–diffusion processes[J]. Nature Machine Intelligence, 2023, 5(7): 765-779.

---

> > > > ### Author Response · Authors · 2023-11-21
> > > > **Response to "low number of parameters" concern (2).**
> > > >
> > > > ## 1. Data-driven models need more parameters to fit the data.
> > > > ## **2. The more physics, the less parameters.**
> > > > ## 3. Physics-aware models vs. Data-driven models
> > > >
> > > > As shown in **Table 3** and **Table 4**, the conclusions of PAPM in each comparison experiment with five datasets and two different task settings are consistent with the view of the above review paper, i.t., **when the amount of data is fixed, the more physical information is introduced into the neural network model, the fewer parameters are required for the model.** On the other hand, models with many parameters (e.g., U-FNet and CNO, which have much more than 1M parameters) perform less than PAPM.
> > > >
> > > > **Table 3**: Main results $\epsilon$ across different datasets in time extrapolation task.
> > > >
> > > > | **Config**      | **Burgers2d C Int.** | **Burgers2d C Ext.** | **RD2d C Int.** | **NS2d C Int.** | **Lid2d C Ext.** | **NSM2d C Int.** | **NSM2d C Ext.** |
> > > > |-----------------|---------------------:|---------------------:|----------------:|--------------------:|-----------------:|-----------------:|-----------------:|
> > > > | ConvLSTM   |                0.314 |                0.551 |           0.815 |               0.781 |                1.323 |             0.910 |            1.102 |
> > > > | Dil-ResNet   |                0.071 |                0.136 |           0.021 |               0.152 |               0.261 |             0.288 |            0.314 |
> > > > | time-FNO2D  |                0.173 |                0.233 |           0.333 |               0.118 |                 0.265 |             0.341 |            0.443 |
> > > > | MIONet      |                0.181 |                0.212 |           0.247 |               0.139 |                0.221 |             0.268 |            0.440 |
> > > > | *U-FNet*    |             *0.109*  |             *0.433*  |           *0.239* |           *0.191* |              *0.192* |           *0.257* |         *0.457* |
> > > > | *CNO*      |             *0.112*  |             *0.126*  |           *0.258* |           *0.125* |                   *0.218* |            *0.197* |        *0.355* |
> > > > | PeRCNN       |                0.212 |                0.282 |           0.773 |               0.571 |                      0.534  |             0.493 |           0.493 |
> > > > | PPNN         |                0.047 |                0.132 |           0.030 |               0.365 |                       0.163  |             0.206 |           0.264 |
> > > > | **PAPM (Our)**  |         **0.039** |            **0.101** |         **0.018** |           **0.110** |              **0.160** |           **0.189** |    **0.245** |
> > > >
> > > >
> > > > **Table 4**: Comparison of the number of trainable parameters ($N_P$) across different datasets.
> > > >
> > > > | **Config**      | **Burgers2d C Int.** | **Burgers2d C Ext.** | **RD2d C Int.** | **NS2d C Int.** | **Lid2d C Ext.** | **NSM2d C Int.** | **NSM2d C Ext.** |
> > > > |-----------------|---------------------:|---------------------:|----------------:|--------------------:|-----------------:|-----------------:|-----------------:|
> > > > | ConvLSTM  | 0.175M     | 0.175M     |0.175M     | 0.139M     | 0.211M    |0.211M    |0.211M    |
> > > > | Dil-ResNet| 0.150M     | 0.150M     | 0.150M     | 0.148M     | 0.152M    | 0.152M    | 0.152M    |
> > > > | time-FNO2D| 0.464M     | 0.464M     | 0.464M     | 0.463M     | 0.464M    |0.464M    |0.464M    |
> > > > | MIONet    | 0.261M     | 0.261M     | 0.261M    | 0.261M    | 0.261M    | 0.261M    | 0.261M    |
> > > > | U-FNet  | 9.853M   | 9.853M   | 9.853M   | 9.851M   | 9.854M  |9.854M  |9.854M  |
> > > > | CNO     | 2.606M   | 2.606M   | 2.606M   | 2.600M   | 2.612M  |2.612M  |2.612M  |
> > > > | PeRCNN    | **0.001**M | **0.001**M | **0.001**M| **0.001**M| **0.001**M| **0.001**M| **0.001**M|
> > > > | PPNN      | 1.201M     | 1.201M     | 1.201M     | 1.190M     | 1.213M    |1.213M    |1.213M    |
> > > > | **PAPM**  | 0.014M     | 0.014M     | 0.014M     | 0.034M     | 0.035M    |0.035M    |0.035M    |0.035M    |
> > > >
> > > > ## 4. Scaling Up Parameters
> > > >
> > > > As shown in **Table 5**, we scaled up the parameters of our model (adjusting the hidden_channels of PAPM) on complex plasma dataset in the **Appendix.8**. We found that as the number of parameters increased, the model effect did not significantly improve. So we chose PAPM-16 with less computational overhead as our model. This reinforces the point that the more physics we introduce into our neural network model, the fewer parameters it needs.
> > > >
> > > > **Table 5: Main results ($\epsilon$) and the number of trainable parameters ($N_P$) on Plasma2d.**
> > > > | **Config**    | **$\epsilon$** | **$N_p$**  |
> > > > | ------------- | ---------------| -----------|
> > > > | convLSTM    | 0.750            | 0.080M     |
> > > > | Dil-ResNet   | 0.316           | 0.152M     |
> > > > | timeFNO2d   | 0.286            | 0.465M     |
> > > > | U-FNet   | 0.718            | 10.091M       |
> > > > | CNO     | 0.407            | 2.674M         |
> > > > | PPNN       | 0.228            | 1.300M      |
> > > > | **PAPM-16**   | 0.178           | **0.032M**|
> > > > | **PAPM-32**   | 0.176           | 0.082M    |
> > > > | **PAPM-64**   | **0.171**        | 0.245M   |

---

> > > > > ### Author Response · Authors · 2023-11-22
> > > > > **Response to "low number of parameters" concern (3).**
> > > > >
> > > > > Dear reviewer 9hXa,
> > > > >
> > > > > Thank you for your valuable comments.
> > > > >
> > > > > We are wondering if our response and revision have resolved your concerns about "low number of parameters". If our response has addressed your concerns, we would highly appreciate it if you could re-evaluate our work. If you have any additional questions or suggestions, we would be happy to have further discussions.
> > > > >
> > > > > Best regards,
> > > > >
> > > > > authors

---

### Official Review · Reviewer_uLfn · 2023-11-01

**Soundness:** 3 good
**Presentation:** 3 good
**Contribution:** 3 good
**Rating:** 6
**Confidence:** 2

**Summary:**

The paper proposes a novel method, the Physics-Aware Proxy Model (PAPM), aimed at improving the efficiency and accuracy of process systems modeling. PAPM incorporates a portion of prior physical knowledge (including conservation and constitutive relations) into the model and introduces a new Temporal and Spatial Stepping Method (TSSM), which is claimed to enhance the model's applicability and predictive ability. The authors conduct several tests, indicating that PAPM seemingly outperforms existing data-driven and physics-aware models.

**Strengths:**

1. The paper addresses a critical issue in the field of process systems modeling, proposing an innovative solution that combines partial prior mechanistic knowledge with a holistic temporal and spatial stepping method.
2. The PAPM model shows impressive results in terms of both improved performance and reduced computational costs compared to existing methods.
3. The paper is well-structured and the methodology is clearly explained, with extensive validation.

**Weaknesses:**

1. The paper could dive further into limitations of the method.
2. The paper could benefit from a more detailed comparison with existing methods. While the authors compare their method to state-of-the-art models, it would be helpful to see a more detailed analysis of why their method outperforms these existing approaches.

**Questions:**

- How well would the PAPM model perform on process systems with less well-understood or more complex physical principles?
- Could the proposed model be applied to other types of systems beyond process systems?

---

> ### Author Response · Authors · 2023-11-19
> **Response to Reviewer uLfn (1)**
>
> ## 1. Method Limitations
> - **Q**: The paper could dive further into limitations of the method.
> - **A**: Thank you for pointing out the importance of discussing the limitations and potential scalability of PAPM. Our approach centers around two primary considerations.
>     - Firstly, we aim to extend our model to more realistic process systems, particularly those used in industrial simulations. While our current validation has been on standard 2D spatio-temporal dynamic systems with well-defined process model mechanisms, PAPM has demonstrated a superior balance in accuracy, operational efficiency, and generalization capabilities. A notable example is our application of PAPM to plasma, a complex case involving multi-physics field coupling. The results from this plasma experiment exemplify PAPM's strengths in handling more intricate process systems. Moving forward, we plan to explore PAPM's architecture in multi-physics field coupling scenarios, such as fluid-structure and thermal fluid-structure coupling problems.
>     - The second consideration is the potential scalability of PAPM concepts to a larger model framework. Since most dynamic systems adhere to three primary aspects – convection (with advection being a specific case), diffusion, and source terms – we are keen to integrate this structured design into developing larger-scale models. We aim to construct a foundational model for process systems that can efficiently and accurately model various processes with a unified approach.
>
> ## 2. Comparison with Existing Methods:
> - **Q**: The paper could benefit from a more detailed comparison with existing methods. While the authors compare their method to state-of-the-art models, it would be helpful to see a more detailed analysis of why their method outperforms these existing approaches.
> - **A**: To further demonstrate PAPM's effectiveness, we have expanded our experiments.
>     - 1) In the revised version, we introduced two new baseline models: U-FNets [1] and Convolutional Neural Operators (CNO) [2], as demonstrated in **Table 1**(below). These additions enrich our comparative analysis, allowing us to explore PAPM's performance in relation to baseline models in terms of accuracy, parameter count, computational speed, and data efficiency in **Table 2**(below). PAPM exhibits the most balanced trade-off between parameter count and performance among all methods evaluated, from explicit structures (Burgers2d, RD2d) to implicit (NS2d) and more complex hybrid structures (Lid2d, NSM2d). Notably, even though PAPM utilizes only 1% of the parameters employed by the prior leading method, PPNN, it still outperforms it by a large margin. In a nutshell, our model enhances the performance by an average of 6.4% over nine tasks, which affirms PAPM as a versatile and efficient framework suitable for diverse process systems. For the details of the above five datasets, please refer to **Tab. 2,3** and **Fig. 4,5,6** in the revised paper.

---

> > ### Author Response · Authors · 2023-11-19
> > **Response to Reviewer uLfn (2)**
> >
> > ## 2. Comparison with Existing Methods:
> > **Table 1**: Main results $\epsilon$ across different datasets in time extrapolation task.
> > | **Config**      | **Burgers2d C Int.** | **Burgers2d C Ext.** | **RD2d C Int.** | **NS2d $\nu$=1e-3** | **NS2d $\nu$=1e-4** | **NS2d $\nu$=1e-5** | **Lid2d C Ext.** | **NSM2d C Int.** | **NSM2d C Ext.** |
> > |-----------------|---------------------:|---------------------:|----------------:|--------------------:|--------------------:|--------------------:|-----------------:|-----------------:|-----------------:|
> > | ConvLSTM [3]    |                0.314 |                0.551 |           0.815 |               0.781 |               0.877 |               0.788 |            1.323 |             0.910 |            1.102 |
> > | Dil-ResNet [4]  |                0.071 |                0.136 |           0.021 |               0.152 |               0.511 |               0.199 |            0.261 |             0.288 |            0.314 |
> > | time-FNO2D [5]  |                0.173 |                0.233 |           0.333 |               0.118 |              0.100  |               0.033 |            0.265 |             0.341 |            0.443 |
> > | MIONet [6]      |                0.181 |                0.212 |           0.247 |               0.139 |               0.114 |               0.051 |            0.221 |             0.268 |            0.440 |
> > | *U-FNet* [1]    |             *0.109*  |             *0.433*  |           *0.239* |           *0.191* |             *0.190* |             *0.256* |           *0.192* |           *0.257* |         *0.457* |
> > | *CNO* [2]       |             *0.112*  |             *0.126*  |           *0.258* |           *0.125* |             *0.148* |           **0.030** |           *0.218* |            *0.197* |        *0.355* |
> > | PeRCNN [7]      |                0.212 |                0.282 |           0.773 |               0.571 |               0.591 |               0.275 |            0.534  |             0.493 |           0.493 |
> > | PPNN [8]        |                0.047 |                0.132 |           0.030 |               0.365 |               0.357 |               0.046 |            0.163  |             0.206 |           0.264 |
> > | **PAPM (Our)**  |            **0.039** |            **0.101** |         **0.018** |           **0.110** |         **0.097** |               0.034 |          **0.160** |           **0.189** |    **0.245** |
> >
> > **Table 2: Main results ($\epsilon$), FLOPs, and comparison of the number of trainable parameters ($N_P$) on Lid2d.**
> > | **Config** | **FLOPs** | **$N_p$** | **$\epsilon$** |
> > | ---------- | --------- | ----------- | --------------- |
> > | convLSTM   | 327.54M   | 211k        | 1.323           |
> > | Dil-ResNet | 624.00M   | 152k        | 0.261           |
> > | timeFNO2d  | 6.89M     | 464k        | 0.265           |
> > | MIONet     | 6.89M     | 261k        | 0.221           |
> > | U-FNet     | 559.89M   | 9854k       | 0.192           |
> > | CNO        | 835.37M   | 2612k       | 0.218           |
> > | PeRCNN     | 3.44M     | 1k          | 0.534           |
> > | PPNN       | 348.56M   | 1213k       | 0.163           |
> > | **PAPM**   | **1.22M** | **11k**     | **0.160**       |
> >
> > **Table 3**: Main results $\epsilon$ in Burgers2d dataset for time extrapolation task with $[0.5, 1, 2, 4]$ scaling.
> > | **Config**         | **Burgers2d 0.5** | **Burgers2d 1** | **Burgers2d 2** | **Burgers2d 4** |
> > | ------------------ | ----------------: | --------------: | --------------: | --------------: |
> > | ConvLSTM           |             0.480 |           0.314 |           0.339 |           0.287 |
> > | Dil-ResNet         |             0.223 |           0.071 |           0.216 |           0.226 |
> > | time-FNO2D         |             0.303 |           0.173 |           0.170 |           0.171 |
> > | U-FNet             |             0.409 |           0.109 |           0.294 |           0.305 |
> > | CNO                |             0.154 |           0.112 |           0.139 |           0.101 |
> > | PPNN               |             0.263 |           0.047 |           0.242 |           0.246 |
> > | **Vanilla PAPM**   |             0.144 |     **0.039**   |           0.084 |           0.107 |
> > | **PAPM**           |       **0.071**   |     **0.039**   |     **0.043**   |       **0.045** |

---

> ### Author Response · Authors · 2023-11-19
> **Response to Reviewer uLfn (3)**
>
> ## 2. Comparison with Existing Methods:
> - **Q**: The paper could benefit from a more detailed comparison with existing methods. While the authors compare their method to state-of-the-art models, it would be helpful to see a more detailed analysis of why their method outperforms these existing approaches.
> - **A**: To further demonstrate PAPM's effectiveness, we have expanded our experiments.
>
> - 2) We have also included adaptive resolution experiments across different resolutions on three datasets, highlighting PAPM's adaptive resolution capabilities. Incorporating physics-based knowledge into our models is crucial for achieving resolution independence. In **Appendix A.7.3**, we delve into the adaptive resolution capabilities of PAPM and our baseline models. We also discuss potential enhancements (see **Table 3 PAPM**, below) based on the concept of operator learning, like expanding the channel domain through lift and project functions to improve model adaptability. As demonstrated in Tab.1, The burgers2d dataset is used as an example to show the results of different baselines and PAPM under four resolutions (scale = $0.5$, $1$, $2$, $4$, The corresponding resolution is $[32, 64, 128,  256]$). The results highlight that modifications to the PAPM structure have notably enhanced its adaptive resolution capabilities across different resolutions, confirming the effectiveness of this architecture. PAPM demonstrates robust performance in various scaling scenarios,  indicating its resilience to resolution changes. Compared to other physical-aware methods, PAPM exhibits superior adaptive resolution ability. Moreover, the results of purely data-driven approaches,  which lack this adaptive resolution capability, underscore the importance of integrating physics priors for enhanced adaptability to varying resolutions. For more details, please refer to **Appendix A.7.3**.
>
> ## 3. More complex Applications:
> - **Q**: How well would the PAPM model perform on process systems with less well-understood or more complex physical principles?
> - **A**: We explore a more complex example of low-temperature argon plasma discharge, which presents a challenging scenario with non-regular grids and complex mechanisms. We consider a more complex process system, low-temperature argon plasma discharge, as depicted in the revised paper's **Fig. 9**. The mechanism equation of this process is far more complex than Eq. 1 and 2, involving the coupling of multiple physical fields such as the electric field, temperature field, and pressure field. It does not strictly adhere to the definition in Eq. 1. Still, for the physical quantities to be modeled, their overall pattern follows the primary setting in which convection, diffusion, and source terms act together. Therefore, we can still construct a proxy model for plasma through PAPM. At the same time, the simulation grid is with non-regular grids. As demonstrated in **Table 4**(below), our method surpasses others in accuracy, generalization, and efficiency. For A more detailed introduction, please refer to **Appendix A.8**. We will use this example to show the generalization ability of PAPM further. Since this work is still in progress, we only show partial results and will show more experimental results later.
>
> **Table 4: Main results ($\epsilon$), FLOPs, training and inference time cost (iteration/second), and the number of trainable parameters ($N_P$) on Plasma2d.**
> | **Config**    | **\$\epsilon\$** | **FLOPs**  | **Train** | **Test** | **$N_p$**   |
> | ------------- | ---------------- | ---------- | --------- | -------- | ------------- |
> | convLSTM [3]      | 0.750            | 2.05G      | 5.05      | 0.43     | 0.080M        |
> | Dil-ResNet [4]    | 0.316            | 3.90G      | 7.61      | 0.59     | 0.152M        |
> | timeFNO2d [5]     | 0.286            | 0.11G      | 2.37      | 0.31     | 0.465M        |
> | U-FNet [1]        | 0.718            | 5.22G      | 9.38      | 1.49     | 10.091M       |
> | CNO [2]           | 0.407            | 3.50G      | 8.46      | 0.92     | 2.674M        |
> | PPNN [8]          | 0.228            | 3.01G      | 4.60      | 0.63     | 1.300M        |
> | **PAPM**      | 0.178            | 0.07G      | 4.66      | 0.47     | 0.032M        |

---

> > ### Author Response · Authors · 2023-11-19
> > **Response to Reviewer uLfn (4)**
> >
> > ## 4. Extending the Applicability of the Proposed Model to Other System Types
> > - **Q**: Could the proposed model be applied to other types of systems beyond process systems?
> > - **A**: When we encounter a new process model, we need to adapt PAPM to adhere to the structured definition of the new model, as illustrated in the model structure of the revised paper's **Fig. 2**. This is precisely the focus of our recent efforts—to extend PAPM to a broader range of systems. For example, in the case of general dynamics, where equations take the form $\partial_t u = \mathcal{F}(x,t; u,\nabla u,\ldots)$, we are exploring the integration of PAPM in this context.
> >
> > In summary, these enhancements to our experimental section highlight PAPM's unique strengths, especially its balanced trade-off between accuracy and computational efficiency, and its remarkable out-of-sample generalization capabilities.
> >
> > **Reference**
> > - [1] Gupta J K, Brandstetter J. Towards multi-spatiotemporal-scale generalized pde modeling[J]. arXiv preprint arXiv:2209.15616, 2022.
> > - [2] Raonić B, Molinaro R, Rohner T, et al. Convolutional Neural Operators[J]. arXiv preprint arXiv:2302.01178, 2023.
> > - [3] Shi X, Chen Z, Wang H, et al. Convolutional LSTM network: A machine learning approach for precipitation nowcasting[J]. Advances in neural information processing systems, 2015, 28.
> > - [4] Stachenfeld K, Fielding D B, Kochkov D, et al. Learned coarse models for efficient turbulence simulation[J]. arXiv preprint arXiv:2112.15275, 2021.
> > - [5] Li Z, Kovachki N, Azizzadenesheli K, et al. Fourier neural operator for parametric partial differential equations[J]. arXiv preprint arXiv:2010.08895, 2020.
> > - [6] Jin P, Meng S, Lu L. MIONet: Learning multiple-input operators via tensor product[J]. SIAM Journal on Scientific Computing, 2022, 44(6): A3490-A3514.
> > - [7] Rao C, Ren P, Wang Q, et al. Encoding physics to learn reaction–diffusion processes[J]. Nature Machine Intelligence, 2023, 5(7): 765-779.
> > - [8] Liu X Y, Sun H, Zhu M, et al. Predicting parametric spatiotemporal dynamics by multi-resolution PDE structure-preserved deep learning[J]. arXiv preprint arXiv:2205.03990, 2022.

---

### Official Review · Reviewer_hyse · 2023-11-01

**Soundness:** 3 good
**Presentation:** 3 good
**Contribution:** 3 good
**Rating:** 5
**Confidence:** 3

**Summary:**

The paper proposes a way to leverage process systems, which is a key model that can be used to emulate a number of physics models. The authors claim that process models are in general complex and difficult to understand and can also lead to incorrect results. In this paper they propose PAPM (physics-aware proxy model) which has the claimed benefit of including physics priors to accomplish better performance on prediction tasks.

**Strengths:**

1. Paper is mostly well written
2. Experiments are clear

**Weaknesses:**

1. While I appreciate the intuitive explanations, process systems are not defined adequately, and this really impedes assessment of the paper. The terms describing this main concept are vague (abstract, introduction and in section 3), and qualitative. Nevertheless, I hope authors can clarify this in the discussion phase (see questions).
2. It is unclear what is required in training vs. at inference
3. The experiments seem to be run for one setting (no monte-carlo simulations)
4. The experiments only consider classical, highly-structured pdes, it is unclear how the proposed model can be used for real-world settings where the dynamics are unknown and may not follow the underlying assumption of (eq.1)

**Questions:**

### Understanding Process Models:

While the contributions seem important it is difficult to understand what process models are. Following are questions which can help authors identify what the reviewer is struggling with, hopefully to help update the paper for a wider audience.
1. Why are the dynamics/equations of the process model unknown? Isn't it defined by the practitioners?
2. In relation to 1, it seems that authors consider dynamics which take the form of eq.1, while the exact values that these quantities take are unknown? Is this true?
3. How are process models different from the proposed model in relation to eq1 and Fig. 3?

### Understanding PAPM:
4. \lambda is defined as "coefficients" in sec 4.1, but it is unclear how they related to eq 1.
5. During training the quantities, t, \lambda, \Phi_0 etc. are available, but during inference, what all inputs are assumed to be available?
6. What is the impact of missing quantities on training, can the model still learn?
7. The structures in Fig 3 (b and c) are still blackboxes, how do these assist in understanding the system as opposed to a process model?


### Minor/semantics/other comments:
1. Why use TSSM for temporal-spatial modeling method (TSSM), TSMM or TSM is more appropriate?
2. The acronyms DF, CF, IST, and EST can be defined just below eq(1) for clarity.
3. Decomposing pde as spatial and temporal modules has been studied in PIML. It is important to discuss these similarities in the present work; see Seo 2021.


Seo et al. 2021, Physics-aware Spatiotemporal Modules with Auxiliary Tasks for Meta-Learning, IJCAI 2021.

---

> ### Author Response · Authors · 2023-11-19
> **Response to Reviewer hyse (1)**
>
> Thank you for your insightful questions, which have allowed us to clarify key aspects of our work.
>
> ## 1. Understanding Process Models:
> - 1.1 **Q**: Why are the dynamics/equations of the process model unknown? Isn't it defined by the practitioners?
>     - **A**:  Process models, integral to our research, consist of conservation equations governing primary physical quantities (mass, energy, momentum) and constitutive equations that establish relationships between various physical variables. While conservation equations are generally known, constitutive relationships often require empirical determination. Our goal with PAPM is to create a structured model that separately and effectively captures these two equation types. For instance, in a chemical process, conservation equations might dictate how substances react over time, while constitutive equations would define specific reaction rates based on temperature or pressure.
>         - conservation equations
>             $$
>             \begin{aligned}
>             &\frac{\partial \Phi}{\partial t}=-\nabla\cdot\left(J_{C}+J_{D}\right)+q+F\
>             &J_{C}=\Phi(x, t) \cdot v,\quad J_{D}=-D \cdot \nabla \Phi
>             \end{aligned}
>             $$
>         - constitutive equations
>             $$
>             \begin{aligned}
>             & v = v(\Phi),\quad D=\lambda \
>             & q=h_O(\Phi),\quad F=h_F\left(X_F\right)
>             \end{aligned}
>             $$
> - 1.2 **Q**: In relation to 1, it seems that authors consider dynamics which take the form of eq.1, while the exact values that these quantities take are unknown? Is this true?
>     - **A**: No. It's important to clarify that our model primarily focuses on the constitutive relationships between physical quantities, rather than on specific physical quantities themselves. The model inputs – initial conditions, coefficients, external source terms, and time (t) – are known during both training and testing phases. The central unknown aspect in our model is the nature of the relationships between these inputs and how they influence the system's state, which embodies the aforementioned constitutive relationships.
>
> - 1.3 **Q**: How are process models different from the proposed model in relation to eq1 and Fig. 3?
>     - **A**: **Fig. 3** depicts our Temporal-Spatial Stepping Method (TSSM), which is tailored to align with the unique equation characteristics of different process systems by employing stepping schemes through temporal and spatial operations, whether in the physical or spectral space. While strictly adhering to Eq. 1 and Eq. 2, the implementation differs based on the specific context. For explicit structures, such as the Burgers equation, we opt to directly employ convolutional kernels in the physical space to capture system dynamics. However, for implicit structures, such as the Navier-Stokes Equation in vorticity form, we consider mapping operations in the spectral space.
> ## 2. Understanding PAPM:
> - 2.1 **Q**: $\lambda$ is defined as "coefficients" in sec 4.1, but it is unclear how they related to eq 1.
>     - **A**: I've made it clear in the new version that $\lambda$ is defined as "coefficients" as follows:
>         $$
>         \begin{aligned}
>         & v = v(\Phi),\quad D=\lambda \
>         & q=h_O(\Phi),\quad F=h_F\left(X_F\right)
>         \end{aligned}
>         $$
>         where $\boldsymbol{v}$ denotes the velocity of the physical quantity being transmitted, $D$ is the coefficient. Here, $v$, $h_O$, and $h_F$ are the corresponding algebraic mapping, and this part determines whether NN is needed for learning mapping according to the specific problem.
>
> - 2.2 **Q**: During training the quantities, t, \lambda, \Phi_0 etc. are available, but during inference, what all inputs are assumed to be available?
>     - **A**: During training and inference, all inputs are assumed to be available, such as initial conditions $\Phi_0$ and coefficients $\lambda$. However, during training, we work with known variables like coefficients, initial conditions, and time, but the specific mappings of these variables are not provided in the model. This design allows PAPM to learn a mapping from a limited number of observations, enabling it to make accurate long-range predictions in the testing phase.
>
> - 2.3 **Q**: What is the impact of missing quantities on training, can the model still learn?
>     - **A**: When missing some quantities on training, the model also learns. For example, when coefficients are unknown, we can define them as a learnable variable.

---

> > ### Author Response · Authors · 2023-11-19
> > **Response to Reviewer hyse (2)**
> >
> > ## 2. Understanding PAPM:
> > - 2.4 **Q**: The structures in Fig 3 (b and c) are still blackboxes, how do these assist in understanding the system as opposed to a process model?
> >     - **A**: Strictly speaking, it should be represented as a gray box (some are known, some are unknown). For instance, the convection and diffusion terms on the left of **Fig.3 (b and c)** exhibit specific forms, such as $-u\nabla u + \nabla^2 u$, which we recognize. Moreover, we can predefine convolution kernels to guide the model's optimization direction. For instance, we might require the diffusion term's kernel to be strictly symmetric to align with the non-directional nature of diffusion behavior. This approach allows for a deeper understanding of the system's physical implications.
> >     - **A**: When we encounter a new process model, we need to adapt PAPM to adhere to the structured definition of the new model, as illustrated in the model structure of **Fig 2**. This is precisely the focus of our recent efforts—to extend PAPM to a broader range of systems. For example, in the case of general dynamics, where equations take the form $\partial_t u = \mathcal{F}(x,t; u,\nabla u,\ldots)$, we are exploring the integration of PAPM in this context.
> >
> > ## 3. Minor/semantics/other comments
> > - 3.1 **Q**: Why use TSSM for temporal-spatial modeling method (TSSM), TSMM or TSM is more appropriate?
> >     - **A**: It is TSSM, short for **Temporal-Spatial Stepping Method**, which I highlight in the revised paper. By utilizing stepping, we want to highlight the dynamic character of the evolution process in time.
> >
> > - 3.2 **Q**: The acronyms DF, CF, IST, and EST can be defined just below eq(1) for clarity.
> >     - **A**: I have reorganized this part of the content to make it more intuitive and straightforward (**see section 3, page 4**).
> >
> > - 3.3 **Q**: Decomposing pde as spatial and temporal modules has been studied in PIML. It is important to discuss these similarities in the present work; see Seo 2021.
> >     - **A**: This is really a detail that needs to be discussed. I have updated it in the Related Work section of the paper.
> >     - **A**: **Spatial and temporal decomposition.** PiMetaL [1] decomposed modeling into spatial and temporal parts, where the former shares the spatial derivatives in the global task, and the latter learns specific adaptively for individuals. NeuralStagger [2] accelerated the solution of PDEs by spatially and temporally decomposing the original learning tasks into several coarser-resolution subtasks. Moreover, these physics-aware methods just discussed [3,4,5,6] all decomposed the spatial and temporal part, where the spatial part updates the state in each time step, and temporal relations are modeled via Neural ODE [7] or Eular schemes. These decomposition ways significantly reduce data requirements and are more conveniently integrated between prior physics with data-driven models for modeling dynamic systems.
> >
> > ## 4. Experiments and Real-world Applications
> > - **4.1 Experiments**
> >     - 1) **Datasets.** We selected three representative papers from the current PIML field and selected their representative two-dimensional datasets, respectively. Burgers with periodic boundary conditions [1], Fitzhugh-Nagumo RD with no-flow Neumann boundary conditions [2], and incompressible Navier-Stokes equation in vorticity [3]. In addition, we have completed two classical fluid field benchmark data sets using COMSOL simulation software. Lid-driven cavity flow and incompressible Navier-Stokes equations with an additional magnetic field. Through these five datasets, the generalization ability of PAPM is comprehensively demonstrated. For the details of the above five datasets, please refer to **Appendix A.5.1**.
> >     - 2) **Evaluation Protocol.** It is worth noting that, unlike other work in PIML field, such as FNO, we consider the model's generalization ability during long-term evolution and strictly distinguish the time step size between the training set and the test set. That is, the training set can only obtain data within the first ($T'$) step, while the test set needs to infer on the full-time step ($T_{end}$). Where $T'\ll T_{end}$, in this paper the experimental set $T_{end}=100$, and $T'=\frac{T_{end}}{2}$.
> >     - 3) **Experimental setup.** Based on this experimental setup, to account for potential variability due to the partitioning process, each experiment is performed three times (different random seeds), and the final result is derived as the average of these three independent runs. Except for the predefined parameters, the parameters of all models are initialized by Xavier, setting the scaling ratio $c=0.02$.

---

> > > ### Author Response · Authors · 2023-11-19
> > > **Response to Reviewer hyse (3)**
> > >
> > > ## 4. Experiments and Real-world Applications
> > > - 4.1 Experiments
> > > - 4.2 More complex Applications:
> > >     -  We consider a more complex process system, low-temperature argon plasma discharge, as depicted in the revised paper's **Fig. 9**. The mechanism equation of this process is far more complex than Eq. 1 and 2, involving the coupling of multiple physical fields such as the electric field, temperature field, and pressure field. It does not strictly adhere to the definition in Eq. 1. Still, for the physical quantities to be modeled, their overall pattern follows the primary setting in which convection, diffusion, and source terms act together. Therefore, we can still construct a proxy model for plasma through PAPM. At the same time, the simulation grid is with non-regular grids. As demonstrated in **Table 1**(below), our method surpasses others in accuracy, generalization, and efficiency. For A more detailed introduction, please refer to **Appendix A.8**. We will use this example to show the generalization ability of PAPM further. Since this work is still in progress, we only show partial results and will show more experimental results later.
> > >
> > > **Table 1: Main results ($\epsilon$), FLOPs, training and inference time cost (iteration/second), and the number of trainable parameters ($N_P$) on Plasma2d.**
> > > | **Config**    | **\$\epsilon\$** | **FLOPs**  | **Train** | **Test** | **\$N_p\$**   |
> > > | ------------- | ---------------- | ---------- | --------- | -------- | ------------- |
> > > | convLSTM [8]      | 0.750            | 2.05G      | 5.05      | 0.43     | 0.080M        |
> > > | Dil-ResNet [9]    | 0.316            | 3.90G      | 7.61      | 0.59     | 0.152M        |
> > > | timeFNO2d [10]     | 0.286            | 0.11G      | 2.37      | 0.31     | 0.465M        |
> > > | U-FNet [11]        | 0.718            | 5.22G      | 9.38      | 1.49     | 10.091M       |
> > > | CNO [12]           | 0.407            | 3.50G      | 8.46      | 0.92     | 2.674M        |
> > > | PPNN [4]          | 0.228            | 3.01G      | 4.60      | 0.63     | 1.300M        |
> > > | **PAPM**      | 0.178            | 0.07G      | 4.66      | 0.47     | 0.032M        |
> > >
> > > Your questions have helped us to better articulate the nuances of our work, and we hope these responses address your queries satisfactorily.
> > >
> > > ## Reference
> > > - [1] Seo S, Meng C, Rambhatla S, et al. Physics-aware spatiotemporal modules with auxiliary tasks for meta-learning[J]. IJCAI 2021.
> > > - [2] Huang X, Shi W, Meng Q, et al. NeuralStagger: accelerating physics-constrained neural PDE solver with spatial-temporal decomposition[J]. arXiv preprint arXiv:2302.10255, 2023.
> > > - [3] Karlbauer M, Praditia T, Otte S, et al. Composing partial differential equations with physics-aware neural networks[C]//International Conference on Machine Learning. PMLR, 2022: 10773-10801.
> > > - [4] Liu X Y, Sun H, Zhu M, et al. Predicting parametric spatiotemporal dynamics by multi-resolution PDE structure-preserved deep learning[J]. arXiv preprint arXiv:2205.03990, 2022.
> > > - [5] Akhare D, Luo T, Wang J X. Physics-integrated neural differentiable (PiNDiff) model for composites manufacturing[J]. Computer Methods in Applied Mechanics and Engineering, 2023, 406: 115902.
> > > - [6] Rao C, Ren P, Wang Q, et al. Encoding physics to learn reaction–diffusion processes[J]. Nature Machine Intelligence, 2023, 5(7): 765-779.
> > > - [7] Chen R T Q, Rubanova Y, Bettencourt J, et al. Neural ordinary differential equations[J]. Advances in neural information processing systems, 2018, 31.
> > > - [8] Shi X, Chen Z, Wang H, et al. Convolutional LSTM network: A machine learning approach for precipitation nowcasting[J]. Advances in neural information processing systems, 2015, 28.
> > > - [9] Stachenfeld K, Fielding D B, Kochkov D, et al. Learned coarse models for efficient turbulence simulation[J]. arXiv preprint arXiv:2112.15275, 2021.
> > > - [10] Li Z, Kovachki N, Azizzadenesheli K, et al. Fourier neural operator for parametric partial differential equations[J]. arXiv preprint arXiv:2010.08895, 2020.
> > > - [11] Gupta J K, Brandstetter J. Towards multi-spatiotemporal-scale generalized pde modeling[J]. arXiv preprint arXiv:2209.15616, 2022.
> > > - [12] Raonić B, Molinaro R, Rohner T, et al. Convolutional Neural Operators[J]. arXiv preprint arXiv:2302.01178, 2023.

---

> > > > ### Author Response · Authors · 2023-11-22
> > > > **A Gentle Reminder**
> > > >
> > > > Dear reviewer hyse,
> > > >
> > > > Thank you for your valuable comments. We have revised our manuscript according to your suggestions.
> > > >
> > > > As we are getting closer to the end of the discussion phase, please let us know if we have appropriately addressed some of your concerns in our rebuttal, like the understanding of process models and PAPM. If you have any additional questions or suggestions, we would be happy to have further discussions.
> > > >
> > > > Best regards,
> > > >
> > > > authors

---

### Official Review · Reviewer_La8N · 2023-11-06

**Soundness:** 3 good
**Presentation:** 3 good
**Contribution:** 3 good
**Rating:** 6
**Confidence:** 2

**Summary:**

This paper proposes a specific structure to encode physics prior to the training and use Euler/RK for time stepping to achieve good generalization capability under a data-scarce situation.

**Strengths:**

The paper explicitly takes into account the physics of the system when designing the system, yielding better generalization capability compare to baselines like FNO

**Weaknesses:**

I am a bit confused with the experimental setting. I really like the argument of baking more physics prior to the model. However, it seems that during the training, the model is still trained with a large-scale dataset - where one needs up to 10^6 times to generate this dataset.

**Questions:**

1. I am curious any thoughts on why FNO performs so badly even with the full dataset for training? This is different from what I generally get from various literature.
2.  I am curious why different padding strategy corresponds to boundary condition. How does it help enforce the boundary condition?
3. how could it generalize to mesh base simulation with adaptive resolution?

---

> ### Author Response · Authors · 2023-11-19
> **Response to Reviewer La8N (1)**
>
> Thank you for your insightful queries regarding our experiment and model performance. Your questions have allowed us to clarify further and highlight the strengths of our approach.
>
> ## 1. The Experimental Setting
> - 1.1. **Datasets.** We selected three representative papers from the current PIML field and selected their representative two-dimensional datasets, respectively. Burgers with periodic boundary conditions [1], Fitzhugh-Nagumo RD with no-flow Neumann boundary conditions [2], and incompressible Navier-Stokes equation in vorticity [3]. In addition, we have completed two classical fluid field benchmark data sets using COMSOL simulation software. Lid-driven cavity flow and incompressible Navier-Stokes equations with an additional magnetic field. Through these five datasets, the generalization ability of PAPM is comprehensively demonstrated. For the details of the above five datasets, please refer to **Appendix A.5.1**.
> - 1.2  **Evaluation Protocol.** It is worth noting that, unlike other work in PIML field, such as FNO, we consider the model's generalization ability during long-term evolution and strictly distinguish the time step size between the training set and the test set. That is, the training set can only obtain data within the first ($T'$) step, while the test set needs to infer on the full-time step ($T_{end}$). Where $T'\ll T_{end}$, in this paper the experimental set $T_{end}=100$, and $T'=\frac{T_{end}}{2}$.
> - 1.3 **Detailed Data Generation Process.** Based on this experimental setting, when simulating through COMSOL, we set the simulation minimum time unit $\delta t$ as $2\times10^{-2}$ s, collect a slice every 10 steps, and collect 100 slices for each data. 500 simulations are performed for each dataset, each with a completely different condition (i.e., initial conditions and equation coefficients, in this case, Reynolds numbers). The computational intensity was significant: a single run in the lid-driven scenario took 91 seconds on average, while the magnetic stirring case took 226 seconds.  The total computation time was approximately $10^6$ s for all 500 cases, highlighting the time-consuming nature of such simulations. The intricacy of multi-physics coupling and the extensive computational demand in these simulations point towards the necessity of more efficient methods.
>
> We hope to build long-range datasets that, in the process, incorporate more complex dynamical behavior of the system rather than the steady-state behavior with short time steps, as demonstrated in the FNO paper. Such datasets can better demonstrate the long-range generalization capability of PAPM.
>
> ## 2. FNO Performance
> Addressing your query on FNO, it is a data-driven approach that shows commendable accuracy within its training domain (**i.t., the training and test datasets are under the same time step**) but struggles with time extrapolation [4,5]. As shown in **Fig. 5** in the paper, while FNO demonstrates commendable accuracy within the training domain ($0<T\leq\frac{1}{2}T_{end}$), its performance falters significantly outside of it ($\frac{1}{2}T_{end}\lt T \leq T_{end}$). However, our model, PAPM, through its unique physics embedding, maintains high accuracy and performance even in extrapolation scenarios to address this limitation.
>
> ## 3. Padding Strategy and Boundary Conditions
> The boundary condition is critical for dynamic modeling because it reflects the interaction between the system and the outside world. Four common boundary conditions are considered in detail: Dirichlet, Neumann, Robin, and Periodic in four directions. It has different mathematical expressions for boundary conditions, but we can transform the continuous space into a discrete grid of points.
>
> Here, the Robin boundary condition is taken as an example. If the boundary condition is given as
> $$\alpha \Phi(X,t) + \beta \frac{\partial \Phi(X,t) }{\partial \mathbf{n}} = f(X,t), X\in \partial \Omega,$$
> the discrete form would be
> $$\alpha \Phi_{Mj} + \beta \frac{\Phi_{(M+1)j} -\Phi_{(M-1)j}}{2\times \delta x}=f_{j}.$$
> We can use a padding method in the convolution kernel $$\Phi_{(M+1)j}=\frac{2\times \delta x}{\beta} (f_{j} - \alpha \Phi_{Mj}) + \Phi_{(M-1)j}.$$
> So, the padding method can be used to enforce the boundary condition. A more detailed introduction can be found in **Appendix A.2**.

---

> ### Author Response · Authors · 2023-11-19
> **Response to Reviewer La8N (2)**
>
> ## 4. Generalization to Mesh-Based Simulation with Adaptive Resolution.
>
> We are actively addressing this issue by breaking it down into two key subproblems:
> -   Generalizing to Mesh-Based Simulation: To adapt to non-uniform meshes common in finite element simulations, we propose two approaches:
>     - a. Using bilinear interpolation to convert non-uniform grids to standard grids, minimizing errors especially in dense grid scenarios.
>     - b. Employing a lift mapping function, similar to the geo-FNO [6] strategy, to transform non-uniform meshes into uniform ones.
>
>     In our case study of a low-temperature argon plasma discharge (see **Fig. 9**), we chose the first approach due to the high density of the simulation grid, allowing for effective direct interpolation before training the PAPM model. We explore a more complex example of low-temperature argon plasma discharge, which presents a challenging scenario with non-regular grids and complex mechanisms. The mechanism equation of this process is far more complex than Eq. 1 and 2, involving the coupling of multiple physical fields such as the electric field, temperature field, and pressure field. It does not strictly adhere to the definition in Eq. 1. Still, for the physical quantities to be modeled, their overall pattern follows the primary setting in which convection, diffusion, and source terms act together. Therefore, we can still construct a proxy model for plasma through PAPM. As demonstrated in **Table 1**(below), our method surpasses others in accuracy, generalization, and efficiency. For A more detailed introduction, please refer to **Appendix A.8**. We will use this example to show the generalization ability of PAPM further.
>
> **Table 1: Main results ($\epsilon$), FLOPs, training and inference time cost (iteration/second), and the number of trainable parameters ($N_P$) on Plasma2d.**
> | **Config**    | **\$\epsilon\$** | **FLOPs**  | **Train** | **Test** | **$N_p$**   |
> | ------------- | ---------------- | ---------- | --------- | -------- | ------------|
> | convLSTM [7]  | 0.750            | 2.05G      | 5.05      | 0.43     | 0.080M      |
> | Dil-ResNet [8]| 0.316            | 3.90G      | 7.61      | 0.59     | 0.152M      |
> | timeFNO2d [3]| 0.286            | 0.11G      | 2.37      | 0.31     | 0.465M      |
> | U-FNet [9]   | 0.718            | 5.22G      | 9.38      | 1.49     | 10.091M     |
> | CNO [10]      | 0.407            | 3.50G      | 8.46      | 0.92     | 2.674M      |
> | PPNN [11]      | 0.228            | 3.01G      | 4.60      | 0.63     | 1.300M      |
> | **PAPM**      | 0.178            | 0.07G      | 4.66      | 0.47     | 0.032M      |

---

> > ### Author Response · Authors · 2023-11-19
> > **Response to Reviewer La8N (3)**
> >
> > ## 4. Generalization to Mesh-Based Simulation with Adaptive Resolution.
> > We are actively addressing this issue by breaking it down into two key subproblems:
> > - Generalizing to Mesh-Based Simulation
> > - Adaptive Resolution:
> >     - Incorporating physics-based knowledge into our models is crucial for achieving resolution independence. In **Appendix A.7.3**, we delve into the adaptive resolution capabilities of PAPM and our baseline models. We also discuss potential enhancements (see **Table 2 PAPM** below) based on the concept of operator learning, like expanding the channel domain through lift and project functions to improve model adaptability.
> >     - As demonstrated in Tab.2, the burgers2d dataset is used as an example to show the results of different baselines and PAPM under four resolutions (scale = $0.5$, $1$, $2$, $4$, The corresponding resolution is $[32, 64, 128,  256]$). The results highlight that modifications to the PAPM structure have notably enhanced its adaptive resolution capabilities across different resolutions, confirming the effectiveness of this architecture. PAPM demonstrates robust performance in various scaling scenarios,  indicating its resilience to resolution changes. Compared to other physical-aware methods, PAPM exhibits superior adaptive resolution ability. Moreover, the results of purely data-driven approaches,  which lack this adaptive resolution capability, underscore the importance of integrating physics priors for enhanced adaptability to varying resolutions. For more details, please refer to **Appendix A.7.3**.
> >
> > **Table 2**: Main results $\epsilon$ in Burgers2d dataset for time extrapolation task with \([0.5, 1, 2, 4]\) scaling.
> > | **Config**         | **Burgers2d 0.5** | **Burgers2d 1** | **Burgers2d 2** | **Burgers2d 4** |
> > | ------------------ | ----------------: | --------------: | --------------: | --------------: |
> > | ConvLSTM[7]           |             0.480 |           0.314 |           0.339 |           0.287 |
> > | Dil-ResNet[8]         |             0.223 |           0.071 |           0.216 |           0.226 |
> > | time-FNO2D[3]         |             0.303 |           0.173 |           0.170 |           0.171 |
> > | U-FNet[9]             |             0.409 |           0.109 |           0.294 |           0.305 |
> > | CNO[10]                |             0.154 |           0.112 |           0.139 |           0.101 |
> > | PPNN[11]               |             0.263 |           0.047 |           0.242 |           0.246 |
> > | **Vanilla PAPM**   |             0.144 |     **0.039**   |           0.084 |           0.107 |
> > | **PAPM**           |       **0.071**   |     **0.039**   |     **0.043**   |       **0.045** |
> >
> > We believe these approaches not only address the immediate challenges but also pave the way for future innovations in our field. We hope these explanations address your concerns, and we are open to further discussion or clarification if needed.
> >
> > ## Reference
> > - [1] Huang X, Li Z, Liu H, et al. Learning to simulate partially known spatio-temporal dynamics with trainable difference operators[J]. arXiv preprint arXiv:2307.14395, 2023.
> >
> > - [2] Takamoto M, Praditia T, Leiteritz R, et al. PDEBench: An extensive benchmark for scientific machine learning[J]. Advances in Neural Information Processing Systems, 2022, 35: 1596-1611.
> >
> > - [3] Li Z, Kovachki N B, Azizzadenesheli K, et al. Fourier Neural Operator for Parametric Partial Differential Equations[C]//International Conference on Learning Representations. 2020.
> >
> > - [4] Michałowska K, et al. Neural operator learning for long-time integration in dynamical systems with recurrent neural networks. arXiv 2023.
> >
> > - [5] Huang X, et al. Learning to simulate partially known spatio-temporal dynamics with trainable difference operators. arXiv 2023.
> >
> > - [6] Li Z, Huang D Z, Liu B, et al. Fourier neural operator with learned deformations for pdes on general geometries[J]. arXiv preprint arXiv:2207.05209, 2022.
> >
> > - [7] Shi X, Chen Z, Wang H, et al. Convolutional LSTM network: A machine learning approach for precipitation nowcasting[J]. Advances in neural information processing systems, 2015, 28.
> >
> > - [8] Stachenfeld K, Fielding D B, Kochkov D, et al. Learned coarse models for efficient turbulence simulation[J]. arXiv preprint arXiv:2112.15275, 2021.
> >
> > - [9] Gupta J K, Brandstetter J. Towards multi-spatiotemporal-scale generalized pde modeling[J]. arXiv preprint arXiv:2209.15616, 2022.
> >
> > - [10] Raonić B, Molinaro R, Rohner T, et al. Convolutional Neural Operators[J]. arXiv preprint arXiv:2302.01178, 2023.
> >
> > - [11] Liu X Y, Sun H, Zhu M, et al. Predicting parametric spatiotemporal dynamics by multi-resolution PDE structure-preserved deep learning[J]. arXiv preprint arXiv:2205.03990, 2022.

---

### Author Response · Authors · 2023-11-19
**Overall Summary Response to All Reviewers**

Thank you to all four reviewers for your insightful comments. We are grateful for the positive feedback on our paper and appreciate your recognition of our work in addressing a significant issue with an innovative approach. Your acknowledgment of our improved performance, computational efficiency, and clarity of our writing and experimentation is encouraging. Your constructive feedback has been instrumental in elevating the quality of our manuscript. In response, we have undertaken several significant revisions:

## 1. Summary of Revision Content

- **PAPM Presentation**: We have restructured the Preliminaries and Methodology sections for better clarity. The introduction to the process model and PAPM's structure now includes an intuitive presentation of physics embedding, with Figure 2 corresponding to Equations 1 and 2. The Temporal-Spatial Stepping Method (TSSM) is elaborated in three distinct cases, detailing the method's relationship with data characteristics and its spatial and temporal aspects. Furthermore, Appendix A.4 now includes a detailed pseudo-code of the TSSM algorithm.
- **Experimental Section Enhancements**: To further demonstrate PAPM's effectiveness, we have expanded our experiments. This includes adding U-FNet (Gupta & Brandstetter, 2022) and CNO (Raoni ́c et al., 2023) as baseline models and conducting data efficiency experiments to showcase PAPM's exceptional generalization performance. We have also included adaptive resolution experiments across different resolutions on three datasets, highlighting PAPM's adaptive resolution capabilities. Additionally, we explore a more complex example of low-temperature argon plasma discharge, which presents a challenging scenario with non-regular grids and complex mechanisms.
- **Additional Updates**: The PAPM source code is now available for reproducibility. We have also enhanced the related work section with a discussion on the spatial and temporal modules in PIML. The time-extrapolation experimental tasks are more intuitively organized into two subcases: coefficient interpolation and coefficient extrapolation, providing clarity to our experimental design. We have added detailed descriptions of the data generation process, and the limitations and future directions of PAPM are thoroughly discussed.

## 2. Core Contributions of the Paper
- The primary contribution of PAPM lies in its novel approach to embedding physics knowledge into the neural network model's structural design. This approach, contrasting with the soft constraints typically seen in PINN models, focuses on incorporating boundary conditions, conservation, and constitutive relations, resulting in improved efficiency and out-of-sample generalizability.
- Another significant contribution is the introduction of the Temporal-Spatial Stepping Method (TSSM). This method adapts to the unique equation characteristics of different process systems, enabling stepping schemes in both physical and spectral spaces.
- Unlike typical experiments in the Physics-Informed Machine Learning (PIML) field, our work includes time extrapolation experiments on five two-dimensional, non-trivial benchmarks, encompassing both parameter interpolation and extrapolation. These nine generalization tasks allow for a comprehensive comparison of PAPM with other baselines, assessing model accuracy, efficiency, and adaptive resolution capabilities. Our exploration into complex examples, such as the low-temperature argon plasma discharge with multi-physics field equations on irregular grids, further attests to PAPM's applicability.

Through this analysis, PAPM's structural design and specialized spatio-temporal stepping method demonstrate an impressive balance between accuracy and computational efficiency, along with notable out-of-sample generalization.

## 3. Closing Remarks
We extend our deepest gratitude to all four reviewers for their constructive feedback. Your insights have significantly improved the paper’s clarity, experimental comprehensiveness, and elucidation of PAPM's contributions. We are committed to further refining our work and welcome any additional feedback or questions.

---

> ### Author Response · Authors · 2023-11-21
> **A Gentle Reminder of Rebuttal Feedbacks**
>
> Dear Reviewers,
>
> We understand that chasing down your replies is not our job, and we intend to avoid adding any pressure to your busy schedules.   However, as we are getting closer to the end of the discussion phase, we would appreciate it if you could let us know if we have appropriately addressed some of your concerns in our rebuttal and if anything can be further clarified.
>
> Thank you very much for your attention. Your prompt response will be greatly appreciated and will assist us in meeting the necessary timelines.
>
> Warm regards,
>
> Authors